# Generative diffusion for perceptron problems: statistical physics analysis and efficient algorithms

**Davide Straziota**\*
Department of Computing Sciences
Bocconi University, Milan, Italy

**Elizaveta Demyanenko**\*
Department of Computing Sciences
Bocconi University, Milan, Italy

**Carlo Baldassi**†
Department of Computing Sciences, BIDSA
Bocconi University, Milan, Italy

**Carlo Lucibello**‡
Department of Computing Sciences, BIDSA
Bocconi University, Milan, Italy

## Abstract

We consider random instances of non-convex perceptron problems in the high-dimensional limit of a large number of examples $M$ and weights $N$, with finite load $\alpha = M/N$. We develop a formalism based on replica theory to predict the fundamental limits of efficiently sampling the solution space using generative diffusion algorithms, conjectured to be saturated when the score function is provided by Approximate Message Passing. For the spherical perceptron with negative margin $\kappa$, we find that the uniform distribution over solutions can be efficiently sampled in most of the Replica Symmetric region of the $\alpha$–$\kappa$ plane. In contrast, for binary weights, sampling from the uniform distribution remains intractable. A theoretical analysis of this obstruction leads us to identify a potential $U(s) = -\log(s)$, under which the corresponding tilted distribution becomes efficiently samplable via diffusion. Moreover, we show numerically that an annealing procedure over the shape of this potential yields a fast and robust Markov Chain Monte Carlo algorithm for sampling the solution space of the binary perceptron.

## 1 Introduction

A fundamental challenge in modern machine learning is the ability to efficiently sample from complex probability distributions. This problem is central to many areas, from generative modeling, where methods such as variational autoencoders and autoregressive transformers aim to approximate real-world data distributions, to Bayesian inference, where the goal is to explore posterior distributions over model parameters.

Diffusion models [Sohl-Dickstein et al., 2015, Ho et al., 2020] have recently emerged as a prominent class of generative models, capable of capturing very complex distributions across various domains, with high sample efficiency. These models simulate a stochastic diffusion process in reverse, starting from a simple distribution and arriving at a complex target distribution $p(\boldsymbol{w})$ via a stochastic differential equation, a process called *denoising* (the corresponding noising process being the forward diffusion process). All dependence on $p(\boldsymbol{w})$ is encoded in a drift term determined by a time-dependent *score function*. In many applications, $p(\boldsymbol{w})$ is not explicitly known, but samples from it are available. A neural approximation to the score function driving the generative process is then learned using the denoising score matching objective [Vincent, 2011].

---

\*equal contributions

†carlo.baldassi@unibocconi.it

‡carlo.lucibello@unibocconi.it

39th Conference on Neural Information Processing Systems (NeurIPS 2025).

Here instead we consider settings in which the target probability density is known, but only up to an intractable normalization factor:

$$p(\boldsymbol{w}) = \frac{\psi(\boldsymbol{w})}{Z}. \tag{1}$$

This formulation is ubiquitous in probabilistic modeling, encompassing energy-based models and posterior inference in Bayesian statistics [LeCun et al., 2007, Neal, 2011]. It arises whenever probabilities are defined through unnormalized potentials, such as in Bayesian inference where the posterior is proportional to the product of likelihood and prior, as well as in physical systems, graphical models, and deep energy-based representations. Typical sampling algorithms in this setting belong to the Markov Chain Monte Carlo (MCMC) family Brooks et al. [2011]. These can suffer from slow mixing times, and their theoretical analysis is generally challenging. In contrast, for generative diffusion, the continuous stochastic process can be well approximated by a small number of discrete steps, and the Bayesian structure induced by the noising/denoising process facilitates theoretical analysis. Recent efforts have explored hybrid approaches combining generative diffusion with MCMC methods [Grenioux et al., 2024, Noble et al., 2025, Vargas et al., 2023, 2025].

A fundamental question is whether the score function can be computed (at all times $t$) by a polynomial-time algorithm with access to the unnormalized density $\psi(\boldsymbol{w})$, enabling efficient sampling from $p(\boldsymbol{w})$ via generative diffusion. An answer to this question would, in turn, naturally bound the performance attainable in settings where only samples from the target distribution are available. Moreover, as we will show, an analysis capable of providing insights into the failure modes of generative diffusion can also suggest modifications and inform algorithmic design, even beyond the original scope.

In this work, we develop a theoretical framework to address this problem precisely in the high-dimensional limit, considering random instances of the distribution $p$ itself (i.e., assuming quenched disorder, in statistical physics terminology). Our framework relies on the non-rigorous but extensively validated replica method from spin glass theory [Mézard et al., 1987, Charbonneau et al., 2023]. The sampling algorithm associated with the replica analysis employs Approximate Message Passing (AMP) [Donoho et al., 2009, Barbier et al., 2019] as the score approximator within the diffusion process. AMP is conjectured to be optimal among polynomial-time algorithms for the denoising task involved in computing the score function.

**Diffusion and AMP**    The diffusion flavor that we use is that of Stochastic Localization (SL) [Eldan, 2013], which can be mapped to a standard denoising diffusion process through time and space transformations [Montanari, 2023, El Alaoui and Montanari, 2022]. We call Algorithmic Stochastic Localization (ASL) the algorithmic implementation of SL on a given target distribution where AMP is used to approximate the score function. ASL was introduced in El Alaoui et al. [2022] and proven to produce fair samples from the Sherrington-Kirkpatrick spin glass model for high enough temperature. ASL has also been applied in statistical inference settings, namely spiked matrix models and high-dimensional regression models [Montanari and Wu, 2023, Cui et al., 2024]. Closest to our work is Ghio et al. [2024], where the authors analyze the analogous of ASL within the stochastic interpolant framework [Albergo and Vanden-Eijnden, 2023] using the replica and cavity methods. They provide asymptotic thresholds for sampling by diffusion in several settings spanning statistical physics, statistics, and combinatorial optimization. Their formalism, however, is limited to so-called "planted" ensembles,[1] which allows to avoid dealing with hard-to-compute normalizing factors $Z$ in the denominator. We overcome this limitation by a double application of the replica trick, which significantly broadens the applicability of the approach. It is worth also mentioning Ricci-Tersenghi and Semerjian [2009] for carrying out a similar analysis on the Belief Propagation-guided decimation scheme for random SAT problems.

**Perceptron Problems**    As applications of our formalism, we investigate the performance of sampling through ASL the solution space of random instances of two non-convex perceptron problems: the spherical perceptron with negative margin and the binary weight perceptron. These are arguably the simplest neural network models whose solution space is non-convex and can therefore provide insights into the behavior of more complex multilayer models. The spherical perceptron with negative margin is a non-convex constraint satisfaction problem introduced in [Franz and Parisi, 2016] as a simple model of glassy behavior. The solution space has a complex star-shaped geometry, recently

---

[1]This includes particular non-planted problems that exhibit the "quiet planting" phenomenon and can be analyzed in the planted regime.

investigated in Annesi et al. [2023]. In the binary weights case instead, the analysis of random instances of the problem has a long tradition in statistical physics [Krauth and Mézard, 1989, Engel and Van den Broeck, 2001]. Most solutions under the uniform distribution are conjecturally algorithmically unreachable by a large class of polynomial algorithms due to the overlap gap property [Gamarnik, 2021, Gamarnik et al., 2022], since typical (according to the uniform distribution) solutions are isolated [Huang and Kabashima, 2014, Baldassi et al., 2023]. Hardness of sampling has been rigorously proven in the symmetric binary perceptron case [El Alaoui and Gamarnik, 2025]. Nonetheless, efficient algorithms for finding solutions do exist [Braunstein and Zecchina, 2006, Baldassi et al., 2016a, Abbe et al., 2022], thanks to the presence of a subdominant but algorithmically accessible dense cluster of solutions [Baldassi et al., 2015]. See Barbier et al. [2024] and Barbier [2025] for further explorations of the geometry of the solution space.

The **main contributions** of this paper are the following:

- We introduce a formalism, based on the analysis of time-dependent potential, $\phi_t(q)$ obtained through the replica method, that gives exact thresholds in the high-dimensionality limit for efficient sampling through generative diffusion. Our method extends the one presented in Ghio et al. [2024] beyond planted models by handling unnormalized target densities.

- We show that in the non-convex spherical perceptron model, the uniform distribution can be efficiently sampled by ASL in a large region of the parameters that define the model (the load $\alpha$ and the margin $\kappa$), roughly corresponding to the Replica Symmetric region from replica theory.

- In the case of binary weights perceptron, we show that the uniform distribution cannot be efficiently sampled, consistent with previous analyses. However, our analysis allows us to discover that ASL can sample efficiently from a distribution tilted by a potential $U(s) = -\log(s)$. This provides the first controlled sampler for the solution space of this problem, with an algorithmic threshold of $\alpha_{\text{alg}} \approx 0.65$ for $\kappa = 0$, rather close to the sat/unsat transition at $\alpha_c \approx 0.83$.

- We adapt the newfound potential inside a simple MCMC annealing procedure that provides fast and robust sampling from the tilted target distribution.

The remainder of this paper is structured as follows: Section 2 introduces the Stochastic Localization process. Section 3 presents an asymptotic analysis of SL using the replica formalism. Section 4 discusses the application of ASL to perceptron problems. In Section 5, we provide conclusions and perspectives.

## 2   Preliminaries on Stochastic Localization

In this Section, we introduce the key components of the sampling algorithm based on the Stochastic Localization (SL) process. Given a probability density $p(\boldsymbol{w})$, with $\boldsymbol{w} \in \mathbb{R}^N$, referred to as the *target distribution*, our goal is to generate samples from $p$. We assume $p$ to be known, possibly up to a hard-to-compute normalization factor, and write it as $p(\boldsymbol{w}) = \psi(\boldsymbol{w})/Z$, where we call *partition function* the normalization $Z = \int \mathrm{d}\boldsymbol{w}\, \psi(\boldsymbol{w})$ and we call $\psi(\boldsymbol{w})$ *unnormalized density*.

Given the target density $p$, SL can be defined as a stochastic differential equation (SDE) that goes from time $t = 0$ to $t = +\infty$ for a vector $\mathbf{h}_t \in \mathbb{R}^N$, which we call the time-dependent *field*. The SL's SDE is the analogous of the reverse process SDE in denoising diffusion [Montanari, 2023]. The initial condition is $\mathbf{h}_0 = \mathbf{0}$, and for $t \geq 0$ the SDE reads:

$$\mathrm{d}\mathbf{h}_t = \mathbf{m}_t(\mathbf{h}_t)\, \mathrm{d}t + \mathrm{d}\mathbf{b}_t, \tag{2}$$

where $(\mathbf{b}_t)_{t \geq 0}$ is the standard Wiener process in $N$ dimensions. The drift term $\mathbf{m}_t(\mathbf{h}_t)$ is computed as the expectation

$$\mathbf{m}_t(\mathbf{h}) = \mathbb{E}_{\boldsymbol{w} \sim p_{\mathbf{h},t}}\left[\boldsymbol{w}\right], \tag{3}$$

over what we call the *tilted distribution* $p_{\mathbf{h},t}(\boldsymbol{w})$, obtained by convolving the target distribution with Gaussian noise:

$$p_{\mathbf{h},t}(\boldsymbol{w}) = \frac{\psi(\boldsymbol{w})\, e^{\langle \mathbf{h}, \boldsymbol{w}\rangle - \frac{t}{2}\|\boldsymbol{w}\|^2}}{Z_{\mathbf{h},t}}. \tag{4}$$

The key feature of the SL process is that as $t \to +\infty$ the field diverges and $p_t := p_{\mathbf{h}_t, t}$ peaks around a single configuration $\boldsymbol{w}^*$ that is statistically distributed (over the realizations of the process) as a sample from the target distribution $p$, that is we have

$$p_t \xrightarrow{t \to +\infty} \delta_{\boldsymbol{w}^*}, \quad \boldsymbol{w}^* \sim p. \tag{5}$$

Therefore, a sample from the target distribution $p$ can be obtained as the value of $\mathbf{m}_t(\mathbf{h}_t)$ at large times.

**Bayesian interpretation**    It can be shown [Montanari, 2023] that at any time $t \geq 0$, the solution $\mathbf{h}_t$ of eq. 2 has the same distribution as

$$\mathbf{h}_t = t\boldsymbol{w}^* + \sqrt{t}\mathbf{g}. \tag{6}$$

where $\boldsymbol{w}^* \sim p$ and $\mathbf{g} \sim \mathcal{N}(\mathbf{0}, I_N)$. This is similar to the forward process of denoising diffusion. The distribution $p_{\mathbf{h}_t, t}$ can then be interpreted as the posterior over $\boldsymbol{w}^*$ given the noisy observation $\mathbf{h}_t$. The function $\mathbf{m}_t(\mathbf{h}_t)$ corresponds to the Bayesian denoiser.

**The ASL algorithm**    The Algorithmic Stochastic Localization (ASL) method we use for a given sampling task consists of discretizing time in the SDE eq. 2, running it up to a large but finite final time $t_f$, and decoding the solution from $\mathbf{h}_{t_f}$ or $\mathbf{m}_{t_f}(\mathbf{h}_{t_f})$. The denoiser $\mathbf{m}_t(\mathbf{h}_t)$ is obtained using Approximate Message Passing (AMP) [Donoho and Tanner, 2009], an iterative algorithm that updates the estimated marginal $\mathbf{m}$ until convergence. Therefore, we have a double-loop algorithm in which, at each step of the discretized SDE, the inner AMP loop must be iterated until convergence. While AMP has limited applicability to real-world tasks and datasets due to its failure in handling correlated or low-dimensional data, in the synthetic settings addressed in this paper it provides a fast and efficient algorithm for Bayesian denoising tasks, conjecturally outperforming any other polynomial-time algorithm for large system sizes. Moreover, the fixed points of AMP are in one-to-one correspondence with the stationary points of the replica free entropy discussed in the next section, making the behavior of AMP amenable to simple asymptotic theoretical analysis [Zdeborová and Krzakala, 2016].

## 3    Asymptotic Analysis of ASL with the Replica Formalism

We devise a formalism to investigate the asymptotic behavior of the ASL sampling process in the large system size limit $N \to +\infty$. The formalism relies on the non-rigorous but well-established [Mézard et al., 1987] replica method. The main outcome will be a criterion for the feasibility of fair sampling involving the evaluation of a time-dependent free entropy. What follows is a generalization of the scheme in Ghio et al. [2024] that allows for handling unnormalized densities, such as the ones we will deal with in the next sections.

Given the target distribution $p$, which we assume to be stochastic and drawn from an ensemble of distributions (quenched disorder), the solution $\mathbf{h}_t$ of the SDE eq. 2 (assuming the drift term is correctly estimated) is distributed as $\mathbf{h}_t = t\boldsymbol{w}^* + \sqrt{t}\mathbf{g}$, where $\boldsymbol{w}^*$ is a sample from $p$—referred to as the *reference sample*—and $\mathbf{g}$ is a standard Gaussian noise (see eq. 6). A relevant parameter for tracking the dynamics is the overlap

$$q(t) = \frac{1}{N}\langle \boldsymbol{w}^*, \boldsymbol{w}_t \rangle. \tag{7}$$

We argue that this quantity, while fluctuating over the realization of $p$ and of the SDE path, concentrates for large $N$ to a deterministic quantity. This quantity emerges naturally as an order parameter in the replica computation that we now describe.

In statistical physics, the free entropy is a central quantity because it gives access to key observables and characterizes the typical behavior of the system. In our case, for a given time $t$, we define the asymptotic average free entropy (or just free entropy) as

$$\phi_t = \lim_{N \to +\infty} \frac{1}{N} \mathbb{E} \log Z_{\mathbf{h}_t, t}, \tag{8}$$

where the expectation is computed over the realization of the target unnormalized density $\psi$ over the reference sample $\boldsymbol{w}^*$ (i.e., over the quenched disorder) and over the Gaussian noise $\mathbf{g}$ of eq. 6.

To handle both the (usually) intractable expectation of a logarithm and the sampling from an unnormalized distribution, we employ the replica method twice: we introduce $s$ replicas of the target distribution to account for the normalization (using $Z^{-1} = \lim_{s \to 0} Z^{s-1}$) and $n$ replicas for the tilted distribution to linearize the logarithm (using $\log Z = \lim_{n \to 0} \partial_n Z^n$), and obtain:

$$\phi_t = \lim_{N \to +\infty} \frac{1}{N} \lim_{s \to 0} \lim_{n \to 0} \partial_n \, \mathbb{E}_{\psi, \boldsymbol{g}} \int \prod_{\alpha=1}^{s} \psi \, (\mathrm{d}\boldsymbol{w}_\alpha^*) \prod_{a=1}^{n} \psi \, (\mathrm{d}\boldsymbol{w}_a) \, e^{\langle \mathbf{h}_t, \boldsymbol{w}_a \rangle - \frac{t}{2} \|\boldsymbol{w}_a\|^2}, \tag{9}$$

where $\mathbf{h}_t = t\boldsymbol{w}_1^* + \sqrt{t}\mathbf{g}$ is computed from $\boldsymbol{w}_1^*$, the first ($\alpha = 1$) of the $s$ replicas $\boldsymbol{w}_\alpha^*$ associated to the target distribution, which thus has a special role. The $n$ replicas $\boldsymbol{w}_a$, on the other hand, have symmetric roles. The structure of this computation is closely related to the one presented in the seminal work of Franz and Parisi [1995]. In the dense systems considered here, the resulting expression depends on all pairwise overlaps among the $n + s$ replicas, determined in the large $N$ limit by a saddle point computation. To perform the $n \to 0$ and $s \to 0$ limits, we assume the Replica Symmetric (RS) ansatz [Mézard et al., 1987, Mézard and Montanari, 2009], that is, we restrict the saddle point evaluation to the most symmetric overlap structure under replica exchanges compatible with the symmetries of eq. 9.[2] Moreover, thanks to the Bayesian structure of the problem, the Nishimori conditions [Nishimori, 1980, Iba, 1999, Ghio et al., 2024] can be applied to further simplify the overlap structure by enforcing additional symmetries (see Appendix B.3). Under these hypotheses, the overlap $q(t)$ of eq. 7 can be computed from the overlaps of replicas involved in eq. 9 as

$$q(t) = \frac{1}{N} \langle \boldsymbol{w}_1^*, \boldsymbol{w}_a \rangle = \frac{1}{N} \langle \boldsymbol{w}_a, \boldsymbol{w}_b \rangle, \qquad \forall a, b \in [n] \text{ and } b \neq a. \tag{10}$$

At time $t = 0$, all the $n + s$ replicas are equivalent, therefore $q(t = 0) = r$, where $r$ is the overlap between two distinct samples of the target distribution: $r = \langle \boldsymbol{w}_\alpha^*, \boldsymbol{w}_\beta^* \rangle / N$ for $\alpha, \beta \in [s]$ and $\alpha \neq \beta$. We further mention that while the Nishimori conditions found for the planted distributions analyzed in Ghio et al. [2024] guarantee the correctness of the RS ansatz, in our more general setting they guarantee only that the analysis of the tilted system doesn't involve further replica symmetry breaking (RSB) compared to the standard replica analysis of the reference systems. Therefore, our RS analysis is exact only in the RS region of $p$.

In general, the replica computation will arrive at an expression in which the free entropy $\phi_t$ is expressed as the saddle point of a function of several order parameters, one of which is $q$:

$$\phi_t = \max_q \, \phi_t(q), \qquad \phi_t(q) \coloneqq \underset{\theta}{\mathrm{extr}} \, \phi_t(q, \theta), \tag{11}$$

where we denoted with $\theta$ all the remaining order parameters except $q$. The value $q_{\max}$ that maximizes $\phi_t(q)$ represents the typical value of $q(t)$.

**Success and Failure of ASL** As shown in Ghio et al. [2024], the study of $\phi_t(q)$ can reveal whether the ASL sampling scheme can recover samples from the target distribution, in the large $N$ limit. More specifically, the success of ASL hinges on the unimodality of $\phi_t(q)$ as a function of $q$. If it has a single maximum at all times $t$, moving smoothly from low $q$ to $q = 1$ (assuming $\|\boldsymbol{w}\|^2 = N$), then the AMP messages correctly recover (with high probability in the limit of large $N$) the value $q(t) = \mathrm{argmax}_q \, \phi_t(q)$ and the algorithm successfully samples from the target distribution. Conversely, if $\phi_t(q)$ becomes multimodal, it could be the case that AMP doesn't return the correct estimate of $\mathbf{m}_t(\mathbf{h}_t)$ and the SDE integration fails. The mechanism is as follows: at $t = 0$, there is always a maximum located at low $q$, which is the one found by AMP. As $t$ increases, this maximum will in general move toward higher $q$, and AMP will follow it; however, if at any $t$ there is a second, higher maximum and at higher $q$, it should be the correct one that solves the saddle point equations, but AMP will miss it. As we show below, this can happen both if the global maximum exists from the outset or if it develops over time. See Fig. 1 (Left) for a setting in which a high $q$ maximum first appears and then becomes the global one.

---

[2]This may limit the exactness of our results to certain regions of parameter space, but the methodology can be straightforwardly—if laboriously—extended to replica-symmetry-broken regimes.

# 4 Applications on Perceptron Models

## 4.1 Definitions

The perceptron [Rosenblatt, 1958] is the simplest neural network model, used for binary classification tasks. Instead of the usual optimization perspective, we adopt a constraint satisfaction one [Engel and Van den Broeck, 2001], and define a family of probability distributions over the solution space. The problem is defined by a dataset $X$, containing $M$ examples $\boldsymbol{x}^\mu \in \mathbb{R}^N$, and a scalar $\kappa$ that we call margin. For simplicity, we assume all labels are equal to 1. A given *weight configuration* $\boldsymbol{w} \in \mathbb{R}^N$ is called a *solution* if all the corresponding stabilities, defined as $s^\mu = \frac{\langle \boldsymbol{x}^\mu, \boldsymbol{w} \rangle}{\sqrt{N}}$, satisfy $s^\mu \geq \kappa \; \forall \mu$. We define a family of distributions over the solution space by the unnormalized density (from eq. 1):

$$\psi(\boldsymbol{w}) = P(\boldsymbol{w}) \prod_{\mu=1}^{M} \Theta\left(s^\mu - \kappa\right) e^{-\frac{1}{T}U(s^\mu - \kappa)}, \quad \text{with} \;\; s^\mu = \frac{\langle \boldsymbol{x}^\mu, \boldsymbol{w} \rangle}{\sqrt{N}} \tag{12}$$

Here $\Theta(s)$ is the Heaviside step function, $\Theta(s) = 1$ if $s > 0$ and 0 otherwise. The parameter $T \geq 0$ is called temperature, and we call *potential* the function $U(s)$. Notice that for $U(s) = 0$ (or any constant) or for $T \to +\infty$, the distribution $p(\boldsymbol{w}) = \psi(\boldsymbol{w})/Z$ becomes the uniform distribution over the solution space. The distribution $P(\boldsymbol{w})$ is a prior on the weights, possibly unnormalized. In the paper, we consider two different priors, corresponding to the spherical weights perceptron, $P(\boldsymbol{w}) = \delta(\|\boldsymbol{w}\|^2 - N)$, and to the binary weights perceptron, $P(\boldsymbol{w}) = \prod_i P(w_i)$ and $P(w_i) = \delta(w_i - 1) + \delta(w_i + 1)$.

The perceptron problems are generated by considering i.i.d. examples $\boldsymbol{x}^\mu \sim \mathcal{N}(0, I_N)$ or $\boldsymbol{x}^\mu \sim \text{Unif}(\{-1, +1\}^N)$. The two settings are equivalent for our asymptotic analysis. The high-dimensional limit is obtained for $N \to +\infty$ and $M \to +\infty$ with fixed finite load $\alpha = M/N$. This statistical setting has been extensively studied in the statistical physics literature, see Engel and Van den Broeck [2001] and Gabrié et al. [2023] for broad reviews.

## 4.2 Implementation of ASL

In order to adapt the ASL sampling scheme, discussed in Section 2, to the target distributions of the perceptron family as defined in eq. 12, we have to implement the corresponding AMP algorithm. The AMP fixed point provides an approximation to the Bayesian denoiser $\mathbf{m}_t(\mathbf{h}_t)$ to be used to solve the Stochastic Localization SDE. Since eq. 12 can be seen as a specific type of generalized linear model, we can adapt the GAMP algorithm from Rangan [2011] for our purposes. The message passing scheme is reported as Algorithm 1 and discussed in Appendix A.

## 4.3 Spherical Non-Convex Perceptron

In this section, we present the asymptotic analysis of ASL, considering the spherical case $\|\boldsymbol{w}\|^2 = N$ and uniform distribution, i.e. $U(s) = 0$ in eq. 12. In particular, we're interested in the case $\kappa < 0$, since in that case the space of the solutions (if non-empty, i.e. for small enough $\alpha$) is non-convex on the sphere [Franz and Parisi, 2016, Annesi et al., 2023]. As discussed in Section 3, we characterize the sampling behavior by studying the free entropy $\phi_t$ as a function of $q$. The replica calculation yields a decomposition of the free entropy in terms of an energetic and an entropic components (full derivation in Appendix B):

$$\phi_t(q, \hat{q}) = -\frac{1}{2}\left(\hat{r}_d + \hat{q}q\right) + \hat{r}r + \frac{1}{2}tq + \mathcal{G}_S(\hat{q}) + \alpha\,\mathcal{G}_E(q), \tag{13}$$

$$\mathcal{G}_S(\hat{q}) = -\frac{1}{2}\log(\hat{q} - \hat{r}_d) + \frac{1}{2(\hat{q} - \hat{r}_d)}\left(\hat{q} + 2\hat{r}\frac{(\hat{q} - \hat{r})}{(\hat{r} - \hat{r}_d)} + \frac{(\hat{q} - \hat{r})^2}{(\hat{r} - \hat{r}_d)}\left(\frac{\hat{r}}{\hat{r} - \hat{r}_d} + 1\right)\right), \tag{14}$$

$$\mathcal{G}_E(q) = \int Dz D\gamma \frac{\tilde{H}_{1-q}\left(\gamma\sqrt{r} + z\sqrt{q - r}\right)}{\tilde{H}_{1-r}\left(\gamma\sqrt{r}\right)} \log \tilde{H}_{1-q}\left(\gamma\sqrt{r} + z\sqrt{q - r}\right). \tag{15}$$

The function $\tilde{H}$ is defined in the Appendix, eq. 21. The overlaps $q$ and $r$ have the interpretation discussed in Section 3, with $q$ in particular being the overlap between the denoiser prediction at time $t$ and the clean configuration. The parameters $\hat{q}$ and $\hat{r}$ are their Lagrange conjugates, and $\hat{r}_d$ is the conjugate for the norm constraint of the reference.

The parameters for the reference system, $r$, $\hat{r}$ and $\hat{r}_d$, are independent of $t$, $q$ and $\hat{q}$. They are determined upfront, using the saddle point equation obtained from the reference system free entropy (see Appendix B.4.5). Therefore, they play the role of external parameters in the above expressions.

The above expressions allow us to derive $\phi_t(q) = \mathrm{extr}_{\hat{q}}\, \phi_t(q, \hat{q})$ and gain direct insight into whether the SL sampling scheme can recover samples from the target distribution asymptotically. The left panel of Fig. 1 shows $\phi_t(q)$ as a function of $q$ for different values of $t$, with parameters $\alpha = 278$ and $\kappa = -2.5$. Initially, $\phi_t(q)$ exhibits a single global maximum. However, a second local maximum emerges as $t$ increases, eventually becoming the global optimizer of $\phi_t$. As discussed in Section 3, this transition marks the onset of multimodality in the free entropy landscape. The central panel of Fig. 1 presents the phase diagram for the ASL scheme for a fixed margin $\kappa$, delineating the different sampling regimes. For each point in the $t$-$\alpha$ plane, we test for the presence or absence of distinct local maxima by initializing the optimization of $\phi_t(q, \hat{q})$ in eq. 13 at two different values of $q$, low and high, and checking whether the results coincide.

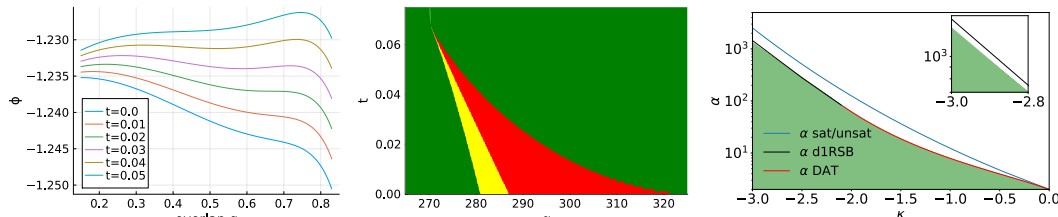

Figure 1: Asymptotic analysis of ASL sampling for the Spherical Perceptron with uniform distribution. **Left:** Free entropy function $\phi_t(q)$ for different values of $t$ and for $\alpha = 278, \kappa = -2.5$. Initially, $\phi_t(q)$ has a single maximum, but as $t$ increases, a second maximum appears, eventually becoming the global one. **Center:** Phase diagram of ASL in the $t$-vs-$\alpha$ plane for $\kappa = -2.5$. *Green region:* $\phi_t(q)$ has a single optimizer, meaning the AMP succeeds at denoising. *Yellow region:* $\phi_t(q)$ has two optimizers, but the global maximum corresponds to the smaller overlap $q$. AMP still succeeds. *Red region:* $\phi_t(q)$ has two optimizers, but the global maximum corresponds to a larger overlap $q$. In this case, AMP fails the denoising task. In order for ASL to succeed at sampling, a vertical line at the corresponding $\alpha$ should lie entirely in the green region. **Right:** Phase diagram delineating the samplable and non-samplable regions for ASL in $\alpha$-vs-$\kappa$ plane. Transition lines predicted from replica theory are taken from Baldassi et al. [2023]. The green region can be sampled by ASL. The zoom in the inset shows the failure of ASL at reaching the d1RSB line.

The right panel of Fig. 1 shows the region in the $\alpha$-$\kappa$ plane where ASL succeeds. The figure also shows replica symmetry-breaking transition lines and the 1RSB prediction for the sat/unsat transition line reported in Baldassi et al. [2023]. Notably, the samplability frontier of ASL coincides with the De Almeida-Thouless (DAT) line and comes slightly short of the dynamical 1-step Replica Symmetry Breaking (1RSB) transition. This result highlights a fundamental limitation: the SL algorithm fails to sample from the target distribution when Replica Symmetry Breaking (RSB) occurs, which signals the fragmentation of the solution space into disconnected clusters. In other words, ASL ceases to function as soon as ergodicity is broken. In the case of the DAT transition, ergodicity is continuously broken and ASL reaches the transition line. In presence of a discontinuous (d1RSB) transition, failure of ASL happens earlier. See Ghio et al. [2024] for more details of phase transitions in similar contexts and Mézard et al. [1987], Mézard and Montanari [2009] for the generic RSB picture.

To validate numerically the uniformity of the ASL sampling, we compare the empirical distribution of stabilities $s^\mu$ obtained via sampling to the asymptotic theoretical prediction, which can also be obtained by the replica method. The derivation and the results are reported in Appendix C. The empirical results match perfectly with the predictions, in the whole region where ASL successfully samples a solution to the constraint satisfaction problem. This strongly suggests that the obtained samples are distributed according to the target distribution, i.e., uniformly over the solution space in this case, as expected.

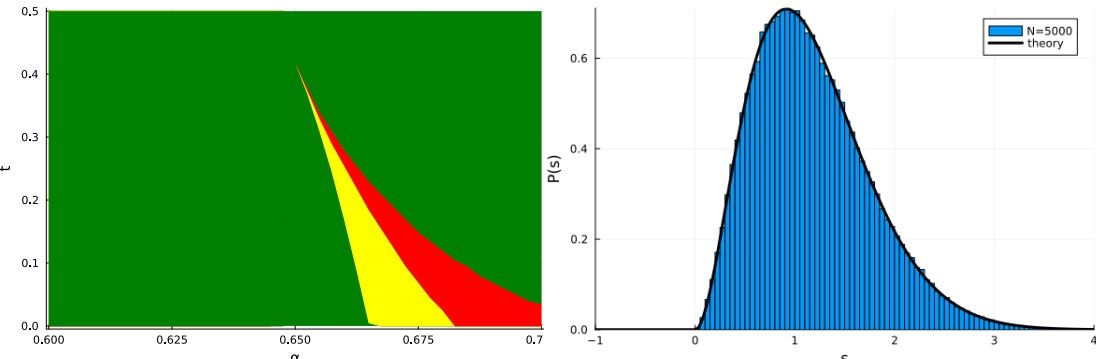

Figure 2: ASL sampling for the Binary Perceptron with $U(s) = -\log(s)$ potential. **Left:** Phase diagram of ASL in the $t$-vs-$\alpha$ plane for $\kappa = 0$ and $T = 0.5$. The color scheme is the same as for the central panel of Figure 1. **Right:** Empirical distribution of the stabilities $s^\mu$ for a configuration obtained by ASL in the case of binary perceptron with the log-potential, $N = 5000$, $\kappa = 0, T = 0.5$, and $\alpha = 0.3$. The black line is the asymptotic theoretical prediction. The excellent agreement shows that ASL produces fair samples from the target distribution.

## 4.4 Binary Perceptron

In this section, we analyze the samplability of the binary perceptron measure. We focus on the zero margin case, $\kappa = 0$. The replica analysis is reported in Appendix E; the final expression for the free entropy $\phi_t(q, \hat{q})$ at time $t$ is the same as eq. 13, except for the entropic term, which becomes

$$\mathcal{G}_S(\hat{q}) = \sum_{w^*=\pm 1} \int Dz D\gamma \, \frac{e^{\gamma\sqrt{\hat{r}}w^*}}{2\cosh(\gamma\sqrt{\hat{r}})} \log\left(2e^{-\frac{1}{2}(\hat{q}-\hat{r}_d)}\cosh\left(z\sqrt{\hat{q}-\hat{r}}+\gamma\sqrt{\hat{r}}+(\hat{q}-\hat{r})w^*\right)\right). \tag{16}$$

### 4.4.1 Selecting a samplable potential

**Sampling fails for non-diverging potentials** As stated in Section 3, ASL fails in the presence of a second peak in the free entropy $\phi_t(q)$ that becomes the global maximum at some time. For the binary perceptron under the uniform distribution, i.e. with $U(s) = 0$, $\phi_t(q)$ exhibits a permanent peak at $q = 1$ with an infinite derivative, for all $\alpha > 0$ and all $t > 0$. This observation is related to the frozen-1RSB nature of the binary perceptron [Krauth and Mézard, 1989], implying that most solutions are isolated, and it is consistent with the known hardness of sampling from the uniform distribution [Huang and Kabashima, 2014, El Alaoui and Gamarnik, 2025]. More surprisingly, this difficulty cannot be removed even by tilting with a potential, unless it diverges at the origin, as we will now discuss.

To understand the origin of the peak at $q = 1$, we compute $\frac{d\phi_t(q)}{dq}$, for $q = 1-\epsilon$, with $\epsilon \ll 1$ [Huang and Kabashima, 2014]. The computations are reported in Appendix F. For the binary case, the expression of the free entropy derivative takes the form:

$$\left.\frac{d\phi_t(q)}{dq}\right|_{q=1-\epsilon} = \frac{1}{2}\log(\epsilon) + \alpha C(\epsilon)\epsilon^{-\frac{1}{2}} + O(1) \tag{17}$$

where $C(\epsilon)$ is a function whose scaling with $\epsilon$ depends on the explicit form of the potential $U(s)$. As it turns out, unless $U(s)$ diverges at 0, $C(\epsilon) = O(1)$ and is always positive, and as a consequence the free entropy unavoidably exhibits a peak at $q = 1$ at all $t > 0$ and all $\alpha > 0$. The second peak becomes dominant at large enough time, before the first peak disappears, leading to the failure of ASL sampling.

**Diffusion with Log-potential** The only way to avoid the peak at $q = 1$ is to let the potential $U(s)$ diverge at the origin. The choice $U(s) = -\log(s)$ is particularly simple to analyze: in that case, $C(\epsilon) = O(\epsilon^{\frac{1}{2T}})$ and therefore the corresponding term $C(\epsilon)\epsilon^{-\frac{1}{2}}$ in eq. 17 becomes negligible for

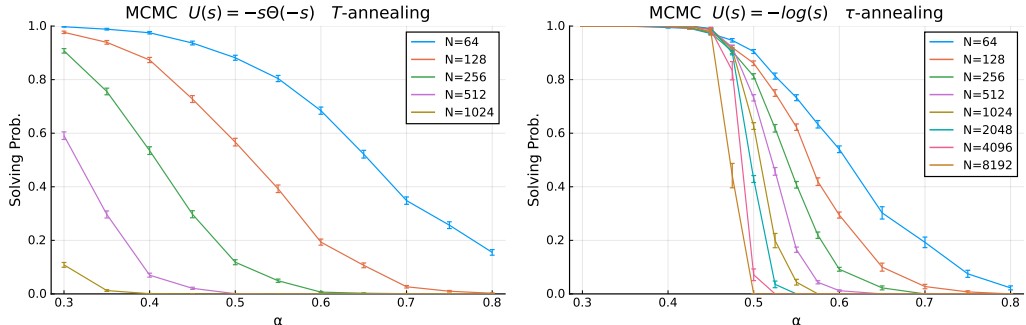

Figure 3: Results for the Binary Perceptron problem, showing the probability of finding a solution as a function of constraint density $\alpha$ and for different system sizes $N$, after 100 sweeps of MCMC. Simulated Annealing on temperature $T$ (left) is compared to our proposed and much more effective $\tau$-annealing scheme.

small $\epsilon$ when $T < 1$. The derivative of the free entropy at $q = 1$ becomes dominated by the logarithmic term and thus $\phi_t(q)$ always has a local minimum at $q = 1$.

In this case, the phenomenology becomes similar to the one observed for the spherical perceptron case, as shown in Figure 2 (Left): with the choice $T = 0.5$, there is a whole range of $\alpha$ up to about $\alpha_{\mathrm{alg}} \approx 0.65$ that is samplable. The transition is close to, although slightly lower than, the best known algorithmic thresholds from heuristic solvers [Braunstein and Zecchina, 2006, Baldassi and Braunstein, 2015, Baldassi et al., 2016a,b], which are known to find solutions from sub-dominant dense clusters. In our case, we sample from the dense cluster as well, but contrary to all solvers we are aware of, the solutions found are fair samples from a target distribution which is fully under control analytically. This is corroborated by the comparison between the empirical stabilities from the ASL sampler and the theoretical predictions, shown in Fig. 2 (Right).

### 4.4.2 $\tau$-annealed MCMC

Using the ASL algorithm in combination with the potential $U(s) = -\log(s)$, we are able to sample solutions of random instances of the binary perceptron problem. ASL, however, inherits the well-known limitations of AMP, generally failing to converge in the presence of structured data, and therefore ASL should not be considered a practical sampling algorithm for generic perceptron problems. On the other hand, direct use of the $\log$-potential in an MCMC algorithm is infeasible, since the potential is not defined on negative stabilities $s$, which means that one should initialize the MC chain from a solution. As a workaround, we propose a reshaping of the potential inspired by the replica trick, which, given a parameter $\tau > 0$, is defined by

$$U_\tau(s) = \begin{cases} \frac{1}{\tau}\left(1 - s^\tau\right) & s > 0, \\ \frac{1}{\tau}\left(1 - s\right) & s \le 0. \end{cases} \tag{18}$$

Notice that $\lim_{\tau \to 0} U_\tau(s) = -\log(s)$ for $s > 0$ and $+\infty$ for $s \le 0$. We set $p_\tau(\mathbf{w}) \propto e^{-\frac{1}{T}\sum_\mu U_\tau(s^\mu)}$ as the moving target density of a Metropolis-Hasting algorithm, where at each MC sweep we decrease linearly $\tau$, starting with initial value 1 and down to 0. Keeping $T$ fixed and with a sufficiently slow annealing, we should be able to sample from the solution space weighted by the $\log$-potential. We call this procedure $\tau$-annealing. In Figure 3, we compare it with a standard Simulated Annealing (SA) on the potential $U(s) = -s\Theta(-s)$, where the temperature $T$ is decreased linearly at each MC sweep from 1 down to 0, so that final samples at $T = 0$ should be distributed according to the uniform measure for a slow enough annealing. While SA fails quickly when increasing $N$, the $\tau$-annealing scheme remains very effective at finding solutions. In the Appendix E.3, we present further experiments showing that, by scaling the number of sweeps as $n_{\mathrm{sweeps}} = N$, we solve up to $\alpha \approx 0.55$ using $\tau$-annealing, while temperature annealing keeps struggling at large system sizes. We also note that, with $\tau$-annealing, the MCMC dynamics is able to diffuse far in solution space, supporting the hypothesis that the algorithm targets a large, dense cluster of solutions. Experiments on real-world datasets reported in E.4, also show improved performance of the $\tau$-annealing scheme compared to other MCMC algorithms.

# 5   Conclusion

In this work, we investigated the feasibility of sampling solutions to perceptron problems via the diffusion scheme based on Approximate Message Passing, known as Algorithmic Stochastic Localization (ASL). For the spherical perceptron, we showed that ASL can sample from the target distribution as long as the free entropy landscape remains unimodal along the trajectory, which holds for constraint density $\alpha = M/N$ below a threshold depending on the margin $\kappa$. In contrast, in the binary case, the uniform distribution is always unsamplable. An investigation of the origin of the issue that prevents ASL from working led us to introduce a potential $U(s) = -\log(s)$ to bias the distribution. This enables efficient sampling over a broad range of $\alpha$ values. This potential also leads to a robust MCMC scheme, $\tau$-annealing, that overcomes ASL's limitations (inherited from AMP) on structured instances of the problem. Looking forward, similar analyses and tailored potentials could enhance sampling and solving in other hard constraint satisfaction problems, especially with isolated solutions. For instance, Budzynski et al. [2019] propose a different reweighting scheme favoring more stable solutions in the context of the k-NAESAT problem. Finally, a promising direction is to rigorously establish our results, particularly for the mathematically simpler binary symmetric perceptron Aubin et al. [2019].

## Acknowledgments

CL acknowledges the European Union - Next Generation EU fund, component M4.C2, investment 1.1 - CUP J53D23001330001, and PRIN PNRR 2022 project P20229PBZR.

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

# Appendix

## A  Approximate Message Passing for the Perceptron

In this Section, we present the Approximate Message Passing (AMP) algorithm used to estimate the drift term $\mathbf{m}_t(\mathbf{h}_t)$ in the Stochastic Localization SDE eq. 2. The AMP algorithm has to be derived for the perceptron target distribution eq. 12, tilted by the observation of $\mathbf{h}_t$, as described in eq. 4. The resulting $p_t(\boldsymbol{w})$ corresponds to a specific case of the Generalized Linear Model family, therefore, the AMP variant we use is adapted from the GAMP algorithm proposed in Rangan [2011].

The AMP framework relies on the definition of two key functions ($\phi_{\text{in}}$ and $\phi_{\text{out}}$), commonly referred to as the input- and output-channel free entropies, given by:

$$\phi_{\text{in}}(A, B) = \log \int P(\mathrm{d}w) \, e^{wB - \frac{1}{2}w^2 A}, \tag{19}$$

$$\phi_{\text{out}}(\omega, V) = \log \int \mathrm{d}s \, \Theta(s - \kappa) \, \frac{e^{-\frac{1}{T}U(s-\kappa) - \frac{(s-\omega)^2}{2V}}}{\sqrt{2\pi V}} = \log \tilde{H}_V(-\omega), \tag{20}$$

where

$$\tilde{H}_a(b) = \int Dz \, \Theta(-b - \kappa + \sqrt{a}z) e^{-\frac{1}{T}U(-b-\kappa+\sqrt{a}z)} \tag{21}$$

with the shorthand notation $Dz = \frac{\mathrm{d}z}{\sqrt{2\pi}} e^{-z^2/2}$. For the case $U(s) = 0$, $\tilde{H}$ simplifies to $\tilde{H}_a(b) = H(-\frac{b+\kappa}{\sqrt{a}})$, with $H(x) = \frac{1}{2}\text{erfc}\left(\frac{x}{\sqrt{2}}\right)$. In the binary weights case $\int P(\mathrm{d}w) = \sum_{w=\pm 1}$. For the

spherical case, the $\|\boldsymbol{w}\|^2 = N$ norm constraint has to be relaxed to a factorized Gaussian prior, $P(w) = e^{-\frac{1}{2}\gamma w^2}$, with $\gamma$ tuned adaptively during the AMP iterations. The relaxed prior is equivalent to the hard one in the limit of large $N$, but AMP can handle only factorized priors. As a consequence, the integral in $\phi_{\text{in}}$ can always be computed in closed form. Furthermore, for some choices of the potential $U(s)$, notably $U(s) = 0$ and $U(s) = -\log(s)$, the integration in $\phi_{\text{out}}$ can also be carried out analytically. Therefore, the AMP used in the settings discussed in the main text is quite fast since it does not contain any integrals.

The full set of the AMP equations is given in Algorithm 1.

---

**Algorithm 1** AMP for ASL on the Perceptron model

---

**Input:** Data $X = \{\boldsymbol{x}^\mu\}_{\mu=1,\ldots,M} \in \mathbb{R}^{M \times N}$, ASL time $t \geq 0$ and field $\boldsymbol{h}_t \in \mathbb{R}^N$, $\epsilon > 0$, $K > 0$.
Set $\boldsymbol{m}^0$, $\boldsymbol{\Delta}^0$, $\boldsymbol{V}^0$, $\boldsymbol{\omega}^0$, $\phi_{\text{out}}$, $\phi_{\text{in}}$, $\boldsymbol{A}^0$, $\boldsymbol{B}^0$, $\Delta_{\text{iter}}$, and $k \leftarrow 0$
**while** not converged on $\boldsymbol{m}^t$ and $\boldsymbol{\Delta}^t$ (number of iters $== K$ or $\Delta_{\text{iter}} < \epsilon$) **do**
  Update mean and variance estimates $\omega_\mu, V_\mu$:

$$V_\mu^k \leftarrow \sum_i \frac{(x_i^\mu)^2}{N} \Delta_i^{k-1} \tag{22}$$

$$\omega_\mu^k \leftarrow \sum_i \frac{x_i^\mu}{\sqrt{N}} m_i^{k-1} - V_\mu^k g_\mu^{k-1} \tag{23}$$

  Update estimates $A_i, B_i, g_\mu$:

$$g_\mu^k \leftarrow \partial_\omega \phi_{\text{out}}\left(\omega_\mu^k, V_\mu^k\right) \tag{24}$$

$$A_i^k \leftarrow -\sum_\mu \frac{(x_i^\mu)^2}{N} \partial_\omega^2 \phi_{\text{out}}\left(\omega_\mu^k, V_\mu^k\right) \tag{25}$$

$$B_i^k \leftarrow \sum_\mu \frac{x_i^\mu}{\sqrt{N}} g_\mu^k + m_i^{k-1} A_i^k \tag{26}$$

  Only for spherical case: enforce norm constraint solving eq. 29 for $\gamma$.
  Update marginals $m_i$ and $\Delta_i$:

$$m_i^k \leftarrow \partial_{B_i} \phi_{\text{in}}\left(A_i^k + t, B_i^k + h_{t,i}\right) \tag{27}$$

$$\Delta_i^k \leftarrow \partial_{B_i}^2 \phi_{\text{in}}\left(A_i^k + t, B_i^k + h_{t,i}\right) \tag{28}$$

  $\Delta_{\text{iter}} \leftarrow \|\boldsymbol{m}^k - \boldsymbol{m}^{k-1}\|^2 / N$ and $k \leftarrow k + 1$
**end while**
**Return:** Converged marginals $\boldsymbol{m}^k$ and $\boldsymbol{\Delta}^k$.

---

In the case of spherical perceptron, the algorithm enforces (in expectation) the norm constraint $\|\boldsymbol{w}\|^2 = N$ by assuming a unnormalized prior $P(w) = e^{-\frac{1}{2}\gamma w^2}$, and then solving for $\gamma$ the equation

$$-2 \sum_{i=1}^N \partial_\gamma \phi_{\text{in}}(A_i^k + t, B_i^k + h_{t,i}) = N, \tag{29}$$

in each AMP iteration using a root-finding algorithm. Notice that $\gamma$ appears implicitely in $\phi_{\text{in}}$ defined in eq. 19.

# B  Replica Computation for ASL on the Perceptron

## B.1  General framework

In this Section, we outline the key steps of the replica calculation used to derive the time-dependent free-entropy $\phi_t$ of the Stochastic Localization process, as outlined in Section 3. The formalism is applied to the general perceptron model given in eq. 12, and then specialized on the spherical and binary case in Appendix D and Appendix E respectively.

We assume $M = \alpha N$ Gaussian-distributed examples, $X = \{\boldsymbol{x}^\mu\}_{\mu=1}^M$, $\boldsymbol{x}^\mu \sim \mathcal{N}(\boldsymbol{0}, I_N)$. The weights will have a factorized prior $P(\boldsymbol{w}) = \prod_i P(w_i)$, where we slightly abused the notation. The case of the non-factorizable global spherical prior will be handled naturally by setting $P(w) = 1$ but then constraining the diagonal overlap instead of optimizing it, as explained later.

We want to compute the free entropy at time $t$ of the model in eq. 12 averaged over the realization of the sample and the noise in the SL process (eq. 6):

$$\phi_t = \lim_{N \to +\infty} \frac{1}{N} \mathbb{E} \log Z_{\mathbf{h}_t, t}. \tag{30}$$

This computation is performed using the replica trick twice (the second one is used to express the expectation over the reference configurations):

$$\mathbb{E} \log Z = \lim_{n \to 0} \partial_n \mathbb{E} Z^n \quad \text{and} \quad Z^{-1} = \lim_{s \to 0} Z^{s-1}.$$

We can thus write

$$\phi_t = \lim_{N \to +\infty} \frac{1}{N} \lim_{s \to 0} \lim_{n \to 0} \partial_n \, \mathbb{E}_{\psi, \boldsymbol{g}} \int \prod_{\alpha=1}^s \psi \, (\mathrm{d}\boldsymbol{w}_\alpha^*) \prod_{a=1}^n \psi \, (\mathrm{d}\boldsymbol{w}_a) \, e^{\langle \mathbf{h}_t, \boldsymbol{w}_a \rangle - \frac{t}{2} \|\boldsymbol{w}_a\|^2}, \tag{31}$$

where $\mathbf{h}_t = t\boldsymbol{w}_1^* + \sqrt{t}\mathbf{g}$, with $\mathbf{g} \sim \mathcal{N}(0, I_N)$. For convenience, we define the replicated partition function

$$Z_t^{n,s} = \int \prod_{\alpha=1}^s \psi \, (\mathrm{d}\boldsymbol{w}_\alpha^*) \prod_{a=1}^n \psi \, (\mathrm{d}\boldsymbol{w}_a) \, e^{\langle \mathbf{h}_t, \boldsymbol{w}_a \rangle - \frac{t}{2} \|\boldsymbol{w}_a\|^2}, \tag{32}$$

For the model in eq. 12, the expectation reads

$$\mathbb{E} Z_t^{n,s} = \mathbb{E}_{X, \mathbf{g}} \int \prod_{\alpha=1}^s P(\mathrm{d}\boldsymbol{w}_\alpha^*) \prod_{a=1}^n P(\mathrm{d}\boldsymbol{w}_a) \prod_{\mu\alpha} \tilde{\Theta} \left( \sum_i \frac{w_{\alpha i}^* x_i^\mu}{\sqrt{N}} \right) \prod_{\mu a} \tilde{\Theta} \left( \sum_i \frac{w_{ai} x_i^\mu}{\sqrt{N}} \right) \times \tag{33}$$

$$\times \, e^{t \sum_{ai} w_{1i}^* w_{ai} + \sqrt{t} \sum_{ai} g_i w_{ai} - \frac{1}{2} t \sum_{ai} w_{ai}^2}. \tag{34}$$

where the realization of $\psi$ depends from the realization of the dataset $X$, and $\tilde{\Theta}$ is defined as

$$\tilde{\Theta}(s) = \Theta(s - \kappa) \, e^{-\frac{1}{T} U(s-\kappa)}. \tag{35}$$

Here $\Theta(s)$ is the Heaviside theta function. Introducing $\lambda_\alpha^\mu = \sum_i \frac{w_{\alpha i}^* x_i^\mu}{\sqrt{N}}$ and $u_a^\mu = \sum_i \frac{w_a x_i^\mu}{\sqrt{N}}$, and their conjugate Lagrange multipliers, we obtain

$$\mathbb{E} Z_t^{n,s} = \mathbb{E}_{X, \mathbf{g}} \int \prod_{\alpha=1}^s P(\mathrm{d}\boldsymbol{w}_\alpha^*) \prod_{a=1}^n P(\mathrm{d}\boldsymbol{w}_a) \prod_{\mu\alpha} \frac{\mathrm{d}\lambda_\alpha^\mu \mathrm{d}\hat{\lambda}_\alpha^\mu}{2\pi} \prod_{\mu a} \frac{\mathrm{d}u_a^\mu \mathrm{d}\hat{u}_a^\mu}{2\pi} \prod_{\mu\alpha} \tilde{\Theta} \, (\lambda_\alpha^\mu) \prod_{\mu a} \tilde{\Theta} \, (u_a^\mu) \tag{36}$$

$$\times \, e^{-i \sum_{\mu\alpha} \hat{\lambda}_\alpha^\mu \lambda_\alpha^\mu - i \sum_{\mu a} \hat{u}_a^\mu u_a^\mu + i \sum_{\mu\alpha i} \hat{\lambda}_\alpha^\mu \frac{w_{\alpha i}^* x_i^\mu}{\sqrt{N}} + i \sum_{\mu ai} \hat{u}_a^\mu \frac{w_{ai} x_i^\mu}{\sqrt{N}} + t \sum_{ai} w_{1i}^* w_{ai} + \sqrt{t} \sum_{ai} g_i w_{ai} - \frac{1}{2} t \sum_{ai} w_{ai}^2}. \tag{37}$$

It is now possible to compute the average over the dataset $X$ and the Gaussian noise $\mathbf{g}$, obtaining

$$\mathbb{E} Z_t^{n,s} = \int \prod_{\alpha=1}^s P(\mathrm{d}\boldsymbol{w}_\alpha^*) \prod_{a=1}^n P(\mathrm{d}\boldsymbol{w}_a) \prod_{\mu\alpha} \frac{\mathrm{d}\lambda_\alpha^\mu \mathrm{d}\hat{\lambda}_\alpha^\mu}{2\pi} \prod_{\mu a} \frac{\mathrm{d}u_a^\mu \mathrm{d}\hat{u}_a^\mu}{2\pi} \prod_{\mu\alpha} \tilde{\Theta} \, (\lambda_\alpha^\mu) \prod_{\mu a} \tilde{\Theta} \, (u_a^\mu) \tag{38}$$

$$\times \, e^{-i \sum_{\mu\alpha} \hat{\lambda}_\alpha^\mu \lambda_\alpha^\mu - i \sum_{\mu a} \hat{u}_a^\mu u_a^\mu + t \sum_{ai} w_{1i}^* w_{ai} + \frac{1}{2} t \sum_i \sum_{ab} w_{ai} w_{bi} - \frac{1}{2} t \sum_{ai} w_{ai}^2} \tag{39}$$

$$\times \, e^{-\frac{1}{2N} \sum_{\mu i} (\sum_{\alpha\beta} \hat{\lambda}_\alpha^\mu \hat{\lambda}_\beta^\mu w_{\alpha i}^* w_{\beta i}^* + \sum_{ab} \hat{u}_a^\mu \hat{u}_b^\mu w_{ai} w_{bi} + 2 \sum_{\alpha a} \hat{\lambda}_\alpha^\mu \hat{u}_a^\mu w_{\alpha i}^* w_{ai})}. \tag{40}$$

We now introduce the overlaps

$$q_{ab} = \frac{1}{N} \sum_i w_{ai} w_{bi}, \qquad r_{\alpha\beta} = \frac{1}{N} \sum_i w_{\alpha i}^* w_{\beta i}^*, \qquad p_{\alpha a} = \frac{1}{N} \sum_i w_{\alpha i}^* w_{ai}, \tag{41}$$

where $q_{ab}$ is the overlap between replicas of the tilted distribution, $r_{\alpha\beta}$ is an overlap in the reference system, and $p_{\alpha a}$ is a cross-overlap between reference and tilted system. We also introduce the conjugate parameters $\hat{q}_{ab}, \hat{r}_{\alpha\beta}, \hat{p}_{\alpha a}$ to enforce their definitions.

The overlaps $q_{aa}$ and $r_{\alpha\alpha}$ will be fixed to 1, since $\|\boldsymbol{w}_b\|^2 = N$ in both the spherical and binary case. For the spherical case, the diagonal multipliers $\hat{q}_{aa}$ will have to be optimized, while their value will be irrelevant in the binary case.

With the introduction of the overlaps, and using the factorization of $P(\boldsymbol{w})$, we can write

$$\mathbb{E}Z_t^{n,s} = \int \prod_{\alpha=1}^{s}\prod_i P(\mathrm{d}w_{\alpha i}^*) \prod_{a=1}^{n} P(w_{ai}) \prod_{\alpha \leq \beta} \mathrm{d}r_{\alpha\beta}\mathrm{d}\hat{r}_{\alpha\beta} \prod_{a \leq b} \mathrm{d}q_{ab}\mathrm{d}\hat{q}_{ab} \prod_{\alpha a} \mathrm{d}p_{\alpha a}\mathrm{d}\hat{p}_{\alpha a} \tag{42}$$

$$\times \, e^{-N\frac{1}{2}\sum_{\alpha\beta}\hat{r}_{\alpha\beta}r_{\alpha\beta}-\frac{1}{2}N\sum_{ab}\hat{q}_{ab}q_{ab}-N\sum_{\alpha a}\hat{p}_{\alpha a}p_{\alpha a}} \tag{43}$$

$$\times \, e^{+\frac{1}{2}\sum_{\alpha\beta}\hat{r}_{\alpha\beta}\sum_i w_{\alpha i}^* w_{\beta i}^*+\frac{1}{2}\sum_{ab}\hat{q}_{ab}\sum_i w_{ai}w_{bi}+\sum_{\alpha a}\hat{p}_{\alpha a}\sum_i w_{\alpha i}^* w_{ai}} \tag{44}$$

$$\times \int \prod_{\mu\alpha} \frac{\mathrm{d}\lambda_\alpha^\mu \mathrm{d}\hat{\lambda}_\alpha^\mu}{2\pi} \prod_{\mu a} \frac{\mathrm{d}u_a^\mu \mathrm{d}\hat{u}_a^\mu}{2\pi} \prod_{\mu\alpha} \tilde{\Theta}(\lambda_\alpha^\mu) \prod_{\mu a} \tilde{\Theta}(u_a^\mu) \tag{45}$$

$$\times \, e^{-i\sum_{\mu\alpha}\hat{\lambda}_\alpha^\mu \lambda_\alpha^\mu - i\sum_{\mu a}\hat{u}_a^\mu u_a^\mu + tN\sum_a p_{1a} + \frac{1}{2}tN\sum_{ab}q_{ab} - \frac{t}{2}N\sum_a q_{aa}} \tag{46}$$

$$\times \, e^{-\frac{1}{2}\sum_\mu (\sum_{\alpha\beta}\hat{\lambda}_\alpha^\mu \hat{\lambda}_\beta^\mu r_{\alpha\beta} + \sum_{ab}\hat{u}_a^\mu \hat{u}_b^\mu q_{ab} + 2\sum_{\alpha a}\hat{\lambda}_\alpha^\mu \hat{u}_a^\mu p_{\alpha a})}. \tag{47}$$

Site-dependent and pattern-dependent terms are now decoupled and factorized. A few more straightforward steps lead to the following expression, amenable to saddle point evaluation:

$$\mathbb{E}Z_t^{n,s} = \int \prod_{\alpha \leq \beta} \mathrm{d}r_{\alpha\beta}\mathrm{d}\hat{r}_{\alpha\beta} \prod_{a \leq b} \mathrm{d}q_{ab}\mathrm{d}\hat{q}_{ab} \prod_{\alpha a} \mathrm{d}p_{\alpha a}\mathrm{d}\hat{p}_{\alpha a} \, e^{N\Phi_t}, \tag{48}$$

where

$$\Phi_t = G_I + G_S + \alpha G_E \tag{49}$$

$$G_I = -\frac{1}{2}\sum_{\alpha\beta}\hat{r}_{\alpha\beta}r_{\alpha\beta} - \frac{1}{2}\sum_{ab}\hat{q}_{ab}q_{ab} - \sum_{\alpha a}\hat{p}_{\alpha a}p_{\alpha a} + t\sum_a p_{1a} + \frac{1}{2}t\sum_{ab}q_{ab} - \frac{t}{2}\sum_a q_{aa}, \tag{50}$$

$$G_S = \log \int \prod_{\alpha=1}^{s} P(\mathrm{d}w_\alpha^*) \prod_{a=1}^{n} P(\mathrm{d}w_a) \, e^{+\frac{1}{2}\sum_{\alpha\beta}\hat{r}_{\alpha\beta}w_\alpha^* w_\beta^* + \frac{1}{2}\sum_{ab}\hat{q}_{ab}w_a w_b + \sum_{\alpha a}\hat{p}_{\alpha a}w_\alpha^* w_a}, \tag{51}$$

$$G_E = \log \int \prod_\alpha \frac{\mathrm{d}\lambda_\alpha \mathrm{d}\hat{\lambda}_\alpha}{2\pi} \prod_a \frac{\mathrm{d}u_a \mathrm{d}\hat{u}_a}{2\pi} \prod_\alpha \tilde{\Theta}(\lambda_\alpha) \prod_a \tilde{\Theta}(u_a)$$
$$\times \, e^{-i\sum_\alpha \hat{\lambda}_\alpha \lambda_\alpha - i\sum_a^n \hat{u}_a u_a - \frac{1}{2}\sum_{\alpha\beta}\hat{\lambda}_\alpha \hat{\lambda}_\beta r_{\alpha\beta} - \frac{1}{2}\sum_{ab}\hat{u}_a \hat{u}_b q_{ab} - \sum_{\alpha a}\hat{\lambda}_\alpha \hat{u}_a p_{\alpha a}}. \tag{52}$$

As it is usually done in the replica method, we will first choose an Ansatz for the structure of the overlaps, so that we can perform analytic continuation and take the small $n$ and $s$ limit, followed by saddle point evaluation.

## B.2 Replica Symmetric Ansatz

The Ansatz (RS) we choose is the Replica Symmetric one. Given the symmetries of the problem, it takes the form

$$q_{ab} = \begin{cases} q_d = 1 & \text{if } a = b, \\ q & \text{if } a \neq b \end{cases}, \qquad r_{\alpha\beta} = \begin{cases} r_d = 1 & \alpha = \beta \\ r & \alpha \neq \beta \end{cases}, \qquad p_{\alpha a} = \begin{cases} p^* & \alpha = 1 \\ p & \alpha \neq 1. \end{cases} \tag{53}$$

The Nishimori conditions discussed in the next Section guarantee that if the RS ansatz is correct for the reference system, e.g. s it is by definition for the spherical perceptron in the RS regime, then the RS ansatz above will give a correct prediction for $\phi_t$. More generally, the correct Ansatz for the tilted system has the same number of steps of symmetry breaking as the correct one for the reference system.

## B.3 Nishimori conditions

Thanks to the Bayesian nature of the problem, the Nishimori conditions apply [Iba, 1999, Nishimori, 1980]. They are a set of identities that we will use to further simplify the overlap structure. We will state the identities in our context and review their derivation.

Let's fix the time $t$, and also fix the realization of the disorder in the distribution $p$ (i.e. the realization of the examples $X$). Assume $\boldsymbol{w}^* \sim p$, and call $\boldsymbol{h} = t\boldsymbol{w}^* + \sqrt{t}\boldsymbol{g}$, with $\boldsymbol{g}$ standard Gaussian, the

observed field (as in eq. 6). The corresponding posterior over $\boldsymbol{w}^*$ is $p_{\boldsymbol{h}_t,t}$ given in equation eq. 4. For convenience we drop the time index and write $\boldsymbol{h} = \boldsymbol{h}_t$ and $p(\boldsymbol{w}|\boldsymbol{h}) = p_{\boldsymbol{h},t}(\boldsymbol{w})$.

Given $K$ i.i.d. samples from the posterior, $\boldsymbol{w}_k \sim p(\boldsymbol{w}|\boldsymbol{h})$, given a continuous bounded test function $f : \mathbb{R}^{K \times N} \to \mathbb{R}$, the following identity (Nishimori identity) holds:

$$\mathbb{E}_{\boldsymbol{w}^*}\mathbb{E}_{\boldsymbol{h}|\boldsymbol{w}^*}\mathbb{E}_{\{\boldsymbol{w}_k\}_{k=1}^K|\boldsymbol{h}} \, f(\boldsymbol{w}^*, \boldsymbol{w}_2, \ldots, \boldsymbol{w}_K) = \mathbb{E}_{\boldsymbol{w}^*}\mathbb{E}_{\boldsymbol{h}|\boldsymbol{w}^*}\mathbb{E}_{\{\boldsymbol{w}_k\}_{k=1}^K|\boldsymbol{h}} \, f(\boldsymbol{w}_1, \boldsymbol{w}_2, \ldots, \boldsymbol{w}_K) \quad (54)$$

In order to show this, fixing $K = 2$ for simplicity, we rewrite the $l.h.s.$ of the previous equation as

$$\mathbb{E} \, f(\boldsymbol{w}^*, \boldsymbol{w}_2) = \int \mathrm{d}\boldsymbol{w}^* \, \mathrm{d}\boldsymbol{h} \, \mathrm{d}\boldsymbol{w}_2 \, P(\boldsymbol{w}^*)P(\boldsymbol{h}|\boldsymbol{w}^*)P(\boldsymbol{w}_2|\boldsymbol{h})f(\boldsymbol{w}^*, \boldsymbol{w}_2) \quad (55)$$

$$= \int \mathrm{d}\boldsymbol{w}^* \, \mathrm{d}\boldsymbol{h} \, \mathrm{d}\boldsymbol{w}_2 \, P(\boldsymbol{h})P(\boldsymbol{w}^*|\boldsymbol{h})P(\boldsymbol{w}_2|\boldsymbol{h})f(\boldsymbol{w}^*, \boldsymbol{w}_2), \quad (56)$$

where we used Bayes' rule. The $r.h.s.$ instead can be rewritten as

$$\mathbb{E} \, f(\boldsymbol{w}_1, \boldsymbol{w}_2) = \int \mathrm{d}\boldsymbol{w}^* \, \mathrm{d}\boldsymbol{h} \, \mathrm{d}\boldsymbol{w}_1 \, \mathrm{d}\boldsymbol{w}_2 \, P(\boldsymbol{w}^*)P(\boldsymbol{h}|\boldsymbol{w}^*)P(\boldsymbol{w}_1|\boldsymbol{h})P(\boldsymbol{w}_2|\boldsymbol{h})f(\boldsymbol{w}_1, \boldsymbol{w}_2) \quad (57)$$

$$= \int \mathrm{d}\boldsymbol{h} \, \mathrm{d}\boldsymbol{w}_1 \, \mathrm{d}\boldsymbol{w}_2 \, P(\boldsymbol{h})P(\boldsymbol{w}_1|\boldsymbol{h})P(\boldsymbol{w}_2|\boldsymbol{h})f(\boldsymbol{w}_1, \boldsymbol{w}_2). \quad (58)$$

The $l.h.s.$ and the $r.h.s.$ of eq. 54 are now clearly shown to be the same, once we rename $\boldsymbol{w}^*$ to $\boldsymbol{w}_1$. We can use eq. 54 to establish identities among the overlaps appearing in the replicated free entropy $\Phi_t$ of eq. 49. In fact, within the RS Ansatz, assuming large $N$ and at saddle point, consider the overlaps

$$p^* = \frac{\langle \boldsymbol{w}_1^*, \boldsymbol{w}_b \rangle}{N} \quad \forall b \in [n]; \qquad q = \frac{\langle \boldsymbol{w}_a, \boldsymbol{w}_b \rangle}{N} \quad \forall a, b \in [n], a \neq b. \quad (59)$$

Since in the replicated partition function $\boldsymbol{w}_a$ and $\boldsymbol{w}_b$ are independent samples of the posterior contiditional on $w_1^*$, the Nishimori identities imply

$$p^* = q. \quad (60)$$

Now consider the overlaps

$$p = \frac{\langle \boldsymbol{w}_\alpha^*, \boldsymbol{w}_a \rangle}{N} \quad \forall a \in [n], \forall \alpha \in [s], \alpha \neq 1; \qquad r = \frac{\langle \boldsymbol{w}_\alpha^*, \boldsymbol{w}_\beta^* \rangle}{N} \quad \forall \alpha, \beta \in [s], \alpha \neq \beta. \quad (61)$$

Since $\boldsymbol{w}_a$ is decoupled from $\boldsymbol{w}_\alpha^*$ when $\alpha \neq 0$, its distribution is the same as a reference configuration when $\boldsymbol{w}_1^*$ is traced out, therefore we have

$$p = r. \quad (62)$$

Similar reasoning can be applied to the conjugate order parameters, so that we obtain:

$$\hat{p}^* = \hat{q}, \quad (63)$$

$$\hat{p} = \hat{r}. \quad (64)$$

These conditions simplify considerably the expressions in the computation of the free entropy. The Nishimori conditions can be also used within Replica Symmetry Breaking (RSB) scenarios for the reference distribution $p$. They guarantee that no further RSB steps are needed in the analysis of the tilted free entropy $\phi_t$.

## B.4 Replica Symmetric Free Entropy

### B.4.1 Entropic term

Let's first look at the entropic term $G_S$ within the RS ansatz:

$$G_S = \log \int \prod_{\alpha=1}^s P(\mathrm{d}w_\alpha^*) \prod_{a=1}^n P(\mathrm{d}w_a) \, e^{+\frac{1}{2}(\hat{r}_d - \hat{r})\sum_\alpha w_\alpha^{*2} + \frac{1}{2}(\hat{r} - \hat{p})(\sum_\alpha w_\alpha^*)^2} \quad (65)$$

$$\times e^{+\frac{1}{2}(\hat{q}_d - \hat{q})\sum_a w_a^2 + \frac{1}{2}(\hat{q} - \hat{p})(\sum_a w_a)^2 + (\hat{p}^* - \hat{p})w_1^* \sum_a w_a + \frac{\hat{p}}{2}(\sum_\alpha w_\alpha^* + \sum_a w_a)^2}. \quad (66)$$

Using the RS ansatz, and the Nishimori conditions, this simplifies to:

$$G_S = \log \int \prod_{\alpha=1}^{s} P(\mathrm{d}w_\alpha^*) \prod_{a=1}^{n} P(\mathrm{d}w_a)\, e^{\frac{1}{2}(\hat{r}_d-\hat{r})\sum_\alpha w_\alpha^{*2}} \tag{67}$$

$$\times\, e^{+\frac{1}{2}(\hat{r}_d-\hat{q})\sum_a w_a^2 + \frac{1}{2}(\hat{q}-\hat{r})(\sum_a w_a)^2 + (\hat{q}-\hat{r})w_1^* \sum_a w_a + \frac{\hat{r}}{2}(\sum_\alpha w_\alpha^* + \sum_a w_a)^2}. \tag{68}$$

Using the Hubbard-Stratonovich transformation twice we can get rid of the quadratic summations like $(\sum_a w_a)^2$ in the exponentials, at the cost of introducing two extra integration variables:

$$G_S = \log \int \prod_{\alpha=1}^{s} P(\mathrm{d}w_\alpha^*) \prod_{a=1}^{n} P(\mathrm{d}w_a) \int DzD\gamma\, e^{\frac{1}{2}(\hat{r}_d-\hat{r})\sum_\alpha w_\alpha^{*2}} \tag{69}$$

$$\times\, e^{\frac{1}{2}(\hat{r}_d-\hat{q})\sum_a w_a^2 + z\sqrt{\hat{q}-\hat{r}}\sum_a w_a + (\hat{q}-\hat{r})w_1^* \sum_a w_a + \gamma\sqrt{\hat{r}}(\sum_\alpha w_\alpha^* + \sum_a w_a)}, \tag{70}$$

where we used $Dz = \mathrm{d}z \frac{e^{-z^2/2}}{\sqrt{2\pi}}$ as a shorthand to denote a gaussian integral. This can now be factorized:

$$G_S = \log \int P(\mathrm{d}w_1^*) \int DzD\gamma\, e^{\frac{1}{2}(\hat{r}_d-\hat{r})w_1^{*2} + \gamma\sqrt{\hat{p}}w_1^*} [\mathcal{Z}(w_1^*,z,\gamma)]^n [\mathcal{Z}^*(\eta,\gamma)]^{s-1}, \tag{71}$$

where

$$\mathcal{Z}(w_1^*,z,\gamma) = \int P(\mathrm{d}w) e^{\frac{1}{2}(\hat{r}_d-\hat{q})w^2 + \left(z\sqrt{\hat{q}-\hat{r}} + \gamma\sqrt{\hat{r}} + (\hat{q}-\hat{r})w_1^*\right)w} \tag{72}$$

$$\mathcal{Z}^*(z,\gamma) = \int P(\mathrm{d}w^*) e^{\frac{1}{2}(\hat{r}_d-\hat{r})w^{*2} + \gamma\sqrt{\hat{r}}w^*} \tag{73}$$

$$\tag{74}$$

The last step of the replica computation consists in performing the limit $n \to 0$ and $s \to 0$, which enables to obtain the final expression of $\phi_t$:

$$\mathcal{G}_S(\hat{q}) = \lim_{s\to 0}\lim_{n\to 0} \partial_n G_S \tag{75}$$

The final expression of the entropic term depends on the prior of the weights. Its functional form is reported in section D and E, respectively for the spherical and binary perceptron cases.

### B.4.2 Energetic term

Let's now focus our attention on the energetic term

$$G_E = \log \int \prod_\alpha \frac{\mathrm{d}\lambda_\alpha \mathrm{d}\hat{\lambda}_\alpha}{2\pi} \prod_a \frac{\mathrm{d}u_a \mathrm{d}\hat{u}_a}{2\pi} \prod_\alpha \tilde{\Theta}(\lambda_\alpha) \prod_a \tilde{\Theta}(u_a) \tag{76}$$

$$\times\, e^{-i\sum_\alpha \hat{\lambda}_\alpha \lambda_\alpha - i\sum_a \hat{u}_a u_a - \frac{1}{2}\sum_{\alpha\beta} \hat{\lambda}_\alpha \hat{\lambda}_\beta r_{\alpha\beta} - \frac{1}{2}\sum_{ab} \hat{u}_a \hat{u}_b q_{ab} - \sum_{\alpha a} \hat{\lambda}_\alpha \hat{u}_a p_{\alpha a}}. \tag{77}$$

After using the RS ansatz over the overlap parameters, the Nishimori conditions, and some manipulations the energetic term simplifies

$$G_E = \log \int \prod_\alpha \frac{\mathrm{d}\lambda_\alpha \mathrm{d}\hat{\lambda}_\alpha}{2\pi} \prod_a \frac{\mathrm{d}u_a \mathrm{d}\hat{u}_a}{2\pi} \prod_\alpha \tilde{\Theta}(\lambda_\alpha) \prod_a \tilde{\Theta}(u_a) \tag{78}$$

$$\times\, e^{-i\sum_\alpha \hat{\lambda}_\alpha \lambda_\alpha - i\sum_a \hat{u}_a u_a - \frac{1}{2}(1-r)\sum_\alpha \hat{\lambda}_\alpha^2 + \frac{q-r}{2}\hat{\lambda}_1^2} \tag{79}$$

$$\times\, e^{-\frac{1}{2}(1-q)\sum_\alpha \hat{u}_\alpha^2 - \frac{r}{2}(\sum_\alpha \hat{\lambda}_\alpha + \sum_a \hat{u}_a)^2 - \frac{q-r}{2}\left(\hat{\lambda}_1 + \sum_a \hat{u}_a\right)^2}. \tag{80}$$

We use again two Hubbard-Stratonovich substitutions to make the integrand factorized in the $a$ and $\alpha$ indices, at the cost of introducing two more gaussian integrals:

$$G_E = \log \int Dz\, D\gamma \prod_\alpha \frac{\mathrm{d}\lambda_\alpha \mathrm{d}\hat{\lambda}_\alpha}{2\pi} \prod_a \frac{\mathrm{d}u_a \mathrm{d}\hat{u}_a}{2\pi} \prod_\alpha \tilde{\Theta}(\lambda_\alpha) \prod_a \tilde{\Theta}(u_a) \tag{81}$$

$$\times\, e^{-i\sum_\alpha \hat{\lambda}_\alpha(\lambda_\alpha + \gamma\sqrt{r}) - i\sum_a \hat{u}_a(u_a + \gamma\sqrt{r} + z\sqrt{q-r}) - \frac{1}{2}(1-r)\sum_\alpha \hat{\lambda}_\alpha^2 + \frac{q-r}{2}\hat{\lambda}_1^2} \tag{82}$$

$$\times\, e^{-\frac{1}{2}(1-q)\sum_\alpha \hat{u}_\alpha^2 - iz\sqrt{q-r}\hat{\lambda}_1}. \tag{83}$$

Finally, collecting the factors and taking the limits for $n \to 0$ and $s \to 0$, we arrive at

$$\mathcal{G}_E(q) = \lim_{s \to 0} \lim_{n \to 0} \partial_n G_E \tag{84}$$

$$= \int Dz D\gamma \frac{\tilde{H}_{1-q}\left(\gamma\sqrt{r} + z\sqrt{q-r}\right)}{\tilde{H}_{1-r}\left(\gamma\sqrt{r}\right)} \log \tilde{H}_{1-q}\left(\gamma\sqrt{r} + z\sqrt{q-r}\right). \tag{85}$$

where

$$\tilde{H}_a(b) = \int Dz\, \Theta(-b - \kappa + \sqrt{a}z)e^{-\frac{1}{T}U(-b-\kappa+\sqrt{a}z)} \tag{86}$$

In the uniform measure case $U(s) = 0$, $\tilde{H}$ simplifies to $\tilde{H}_a(b) = H(-\frac{b+\kappa}{\sqrt{a}})$, with $H(x) = \int_x^{+\infty} Dz = \frac{1}{2}\mathrm{erfc}\left(\frac{x}{\sqrt{2}}\right)$.

### B.4.3 Interaction term

Finally we compute the interaction term

$$G_I = -\frac{1}{2}\sum_{\alpha\beta}\hat{r}_{\alpha\beta}r_{\alpha\beta} - \frac{1}{2}\sum_{ab}\hat{q}_{ab}q_{ab} - \sum_{\alpha a}\hat{p}_{\alpha a}p_{\alpha a} + t\sum_a p_{1a} + \frac{1}{2}t\sum_{ab}q_{ab} - \frac{t}{2}\sum_a q_{aa}, \tag{87}$$

which, after using the RS ansatz, the Nishimori conditions, and taking the limits $n \to 0$ and $s \to 0$, reduces to:

$$\mathcal{G}_I = \lim_{s \to 0} \lim_{n \to 0} \partial_n G_I = -\frac{1}{2}\hat{r}_d - \frac{1}{2}\hat{q}q + \hat{r}r + \frac{1}{2}tq \tag{88}$$

### B.4.4 Saddle Point Equations

The optimization of $\phi_t$ can now be performed using the saddle point method [Mézard et al., 1987]. This amounts at solving the following system of equations

$$\hat{q} = t + 2\alpha \frac{\partial}{\partial q}\mathcal{G}_E(q), \tag{89}$$

$$q = 2\frac{\partial}{\partial \hat{q}}\mathcal{G}_S(\hat{q}). \tag{90}$$

The optimization is usually performed iteratively, starting from an initial guess, inserting it in the right hand side, obtaining updated values, and iterating until convergence.

It is important to point out that the overlap parameters for the reference network are not obtained from the free entropy expression derived above; rather, they need to be determined independently from an analogous (but simpler) free entropy expression that only involves the reference replicas. The procedure is schematically shown in the next section, and the full computation is reported in Engel and Van den Broeck [2001].

### B.4.5 Reference system

The free entropy associated to the reference system is simply that of a standard perceptron problem, which is well known and can be found in Engel and Van den Broeck [2001]. The equations of the overlap parameters depend only on $(r, \hat{r}, \hat{r}_d)$. This results from the Bayesian structure of the problem: the reference probability measure is independent of the tilted one; for this reason, the reference free entropy is independent of $q, \hat{q}$. One can recover the expression of reference system free entropy $\phi^{(r)}(r, \hat{r}, \hat{r}_d)$ fixing $n = 0$ and taking the limit $s \to 0$ of $\Phi_t$, in eq. 49. The RS expression reads:

$$\phi^{(r)}(r, \hat{r}, \hat{r}_d) = \mathcal{G}_I^{(r)}(r, \hat{r}, \hat{r}_d) + \mathcal{G}_S^{(r)}(\hat{r}, \hat{r}_d) + \alpha\mathcal{G}_E^{(r)}(r), \tag{91}$$

$$\mathcal{G}_I^{(r)}(r, \hat{r}, \hat{r}_d) = \frac{1}{2}(r\hat{r} - \hat{r}_d), \tag{92}$$

$$\mathcal{G}_S^{(r)}(\hat{r}, \hat{r}_d) = \log\left(\int P(\mathrm{d}w^*)e^{\frac{1}{2}(\hat{r}_d-\hat{r})w^{*2}+\gamma\sqrt{\hat{r}}w^*}\right), \tag{93}$$

$$\mathcal{G}_E^{(r)}(r) = \int Dz \log \tilde{H}_{1-r}\left(\sqrt{r}z\right). \tag{94}$$

The corresponding saddle point equations, that can be used to determine $r$, $\hat{r}$ and $\hat{r}_d$ as a function of $\alpha$ and $\kappa$, are:

$$r = 1 - 2\frac{\partial}{\partial \hat{r}}\mathcal{G}_S^{(r)}(\hat{r}, \hat{r}_d), \tag{95}$$

$$1 = 1 - 2\frac{\partial}{\partial \hat{r}_d}\mathcal{G}_S^{(r)}(\hat{r}, \hat{r}_d), \tag{96}$$

$$\hat{r} = -2\alpha\frac{\partial}{\partial r}\mathcal{G}_E^{(r)}(r), \tag{97}$$

Our treatment differs from the standard one present in the literature only by the addition of the potential $U(s)$, which however only affects the energetic term and is fully absorbed in the definition of the $\tilde{H}$ function.

## C  Computation of the stability distribution

In this Appendix, we perform the computation of the stability distribution in the perceptron problem with an arbitrary potential $U(s)$. This is a simple generalization to include the potential of a standard calculation that can be performed with the replica trick Engel and Van den Broeck [2001].

Given an $N$-dimensional weight vector $\boldsymbol{w}$, representing a solution of the perceptron problem with a margin $\kappa$, and a pattern $\boldsymbol{x}^\mu$, the stability is defined as $s^\mu = \frac{\langle \boldsymbol{w}, \boldsymbol{x}^\mu \rangle}{\sqrt{N}}$. Our goal is to compute, for a given target stability $s$, the average of $\delta(s - s^\mu)$ over all patterns and all solutions sampled from the target distribution $p$ (eq. 1), and subsequently average over all realizations of the patterns:

$$P(s) = \mathbb{E}_X \mathbb{E}_{\boldsymbol{w}\sim p}\frac{1}{M}\sum_{\mu=1}^{M}\delta(s - s^\mu) \tag{98}$$

Since all patterns are statistically equivalent, we can just compute the average stability of the first pattern:

$$P(s) = \mathbb{E}_X \mathbb{E}_{\boldsymbol{w}\sim p}\,\delta(s - s^1). \tag{99}$$

We now expand the formula using the definition of $p(\boldsymbol{w})$ in eq. 12:

$$P(s) = \mathbb{E}_X \frac{\int P(\mathrm{d}\boldsymbol{w})\prod_{\mu=1}^{M}\tilde{\Theta}(s^\mu)\,\delta(s - s^1)}{\int P(\mathrm{d}\boldsymbol{w})\,\tilde{\Theta}(s^\mu)}, \tag{100}$$

where will also used the definition of $\tilde{\Theta}$ in eq. 35. In order to take the expectation, we introduce $n$ replicas, with $n$ integer at first, and to be sent to zero through analytical continuation, so that we can rewrite the denominator as $Z^{-1} = \lim_{n\to 0} Z^{n-1}$. We give a special role to replica 1, and obtain the formula

$$P(s) = \lim_{n\to 0}\mathbb{E}_X \int \prod_a P(\mathrm{d}\boldsymbol{w}_{a=1}^n)\prod_{\mu,a}\left[\tilde{\Theta}\left(\frac{1}{\sqrt{N}}\langle \boldsymbol{w}_a, \boldsymbol{x}^\mu\rangle\right)\right]\delta\left(s - \frac{1}{\sqrt{N}}\langle \boldsymbol{w}_1, \boldsymbol{x}^1\rangle\right). \tag{101}$$

The computation can then be carried out as usual, leading to the free entropy of the reference system in Section B.4.5, followed by saddle point evaluation. In fact, the observable $\delta\left(s - \frac{1}{\sqrt{N}}\langle \boldsymbol{w}_1, \boldsymbol{x}^1\rangle\right)$ being subdominant does not influence the saddle point. Given the stationary values of the overlaps, one can then obtain the RS expression for the stability distribution, given by

$$P(s) = \tilde{\Theta}(s)\frac{1}{\sqrt{2\pi(1-r)}}\int Dt\,\frac{e^{-\frac{(s-\sqrt{r}t)^2}{2(1-r)}}}{\tilde{H}_{1-r}(\sqrt{r}t)}, \tag{102}$$

where $\tilde{H}$ has been defined in eq. 21.

## D  Spherical Perceptron

In this section, we consider the spherical weights case, in which the prior is $P(\boldsymbol{w}) = \delta(\|\boldsymbol{w}\|^2 - N)$.

### D.1 Free Entropy

We obtain the free energy by specializing the entropic term eq. 51 to the spherical prior, which together with eq. 50 and eq. 52 gives:

$$\phi_t(q,\hat{q}) = \mathcal{G}_I(q,\hat{q}) + \mathcal{G}_S(\hat{q}) + \alpha\mathcal{G}_E(q) \tag{103}$$

$$\mathcal{G}_I = -\frac{1}{2}\hat{r}_d - \frac{1}{2}\hat{q}q + \hat{r}r + \frac{1}{2}tq \tag{104}$$

$$\mathcal{G}_S(\hat{q}) = -\frac{1}{2}\log(\hat{q}-\hat{r}_d) + \frac{1}{2(\hat{q}-\hat{r}_d)}\left(\hat{q} + 2\hat{r}\frac{(\hat{q}-\hat{r})}{(\hat{r}-\hat{r}_d)} + \frac{(\hat{q}-\hat{r})^2}{(\hat{r}-\hat{r}_d)}\left(\frac{\hat{r}}{\hat{r}-\hat{r}_d}+1\right)\right) \tag{105}$$

$$\mathcal{G}_E(q) = \int Dz D\gamma \frac{\tilde{H}_{1-q}\left(\gamma\sqrt{r}+z\sqrt{q-r}\right)}{\tilde{H}_{1-r}\left(\gamma\sqrt{r}\right)}\log\tilde{H}_{1-q}\left(\gamma\sqrt{r}+z\sqrt{q-r}\right) \tag{106}$$

### D.2 Reference system

Once the prior for the perceptron model is specified, the free energy equations for the reference system become the following:

$$\phi^{(r)}(r,\hat{r},\hat{r}_d) = \mathcal{G}_I^{(r)}(r,\hat{r},\hat{r}_d) + \mathcal{G}_S^{(r)}(\hat{r},\hat{r}_d) + \alpha\mathcal{G}_E^{(r)}(r), \tag{107}$$

$$\mathcal{G}_I^{(r)}(r,\hat{r},\hat{r}_d) = \frac{1}{2}(r\hat{r}-\hat{r}_d), \tag{108}$$

$$\mathcal{G}_S^{(r)}(\hat{r},\hat{r}_d) = \frac{1}{2}\left(\log(2\pi) - \log(\hat{r}-\hat{r}_d) + \frac{\hat{r}}{(\hat{r}-\hat{r}_d)}\right), \tag{109}$$

$$\mathcal{G}_E^{(r)}(r) = \int Dz \log\tilde{H}_{1-r}\left(\sqrt{r}z\right). \tag{110}$$

They are obtained by specializing the entropic term for the reference system in eq. 93 to the case of spherical perceptron prior, together with eq. 92 and eq. 94.

### D.3 Stability Distribution

In this section, we present the distribution of stabilities computed through ASL, and we compare them with the theoretical expectation, whose derivation is provided in Section C. The distribution of the stabilities for $U(s) = 0$ is reported in Fig. 4 for $\kappa = -2.1$ and several values of $\alpha \in [5,20,80]$.

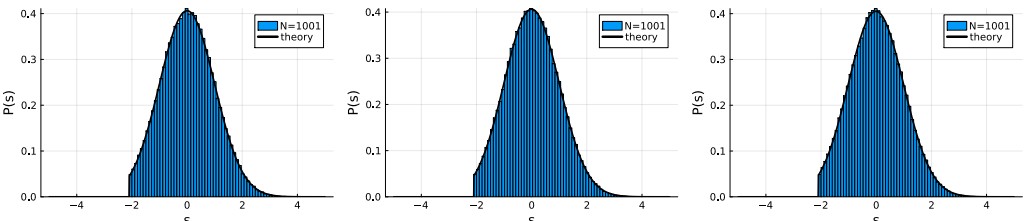

Figure 4: Distribution of the stabilities $s^\mu$ of a spherical perceptron with negative margin, where $s^\mu = \frac{\langle \boldsymbol{w}, \boldsymbol{x}^\mu \rangle}{\sqrt{N}}$. For number of variables $N = 1001$, $\kappa = -2.1$ and $\alpha = 5$ (Left), $\alpha = 20$ (Center) and $\alpha = 80$ (Right). The empirical distribution of the stabilities of the ASL samples (blue) coincides for different parameter regimes with the distribution computed through replica method (black solid line).

## E  Binary Perceptron

In this section, we consider the binary weights case, in which the (improper) prior is $P(w_i) = \delta(w_i - 1) + \delta(w_i + 1)$.

## E.1 Free Entropy

We obtain the free energy by specializing the expression for the entropic term in eq. 51 to the binary prior, together with eq. 50 and eq. 52:

$$\phi_t(q, \hat{q}) = \mathcal{G}_I(q, \hat{q}) + \mathcal{G}_S(\hat{q}) + \alpha \mathcal{G}_E(q), \tag{111}$$

$$\mathcal{G}_S(\hat{q}) = \sum_{w^*=\pm 1} \int Dz \, D\gamma \, \frac{e^{\gamma \sqrt{\hat{r}} w^*}}{2 \cosh(\gamma \sqrt{\hat{r}})} \log \left( 2 e^{\frac{1}{2}(\hat{r}_d - \hat{q})} \cosh \left( z \sqrt{\hat{q} - \hat{r}} + \gamma \sqrt{\hat{r}} + (\hat{q} - \hat{r}) w^* \right) \right), \tag{112}$$

$$\mathcal{G}_E(q) = \int Dz \, D\gamma \, \frac{\tilde{H}_{1-q}\left(\gamma \sqrt{r} + z\sqrt{q-r}\right)}{\tilde{H}_{1-r}\left(\gamma \sqrt{r}\right)} \log \tilde{H}_{1-q}\left(\gamma \sqrt{r} + z\sqrt{q-r}\right), \tag{113}$$

$$\mathcal{G}_I = -\frac{1}{2}\hat{r}_d - \frac{1}{2}\hat{q}q + \hat{r}r + \frac{1}{2}tq. \tag{114}$$

## E.2 Reference system

The free energy equations for the reference system are the following:

$$\phi^{(r)}(r, \hat{r}, \hat{r}_d) = \mathcal{G}_I^{(r)}(r, \hat{r}, \hat{r}_d) + \mathcal{G}_S^{(r)}(\hat{r}, \hat{r}_d) + \alpha \mathcal{G}_E^{(r)}(r), \tag{115}$$

$$\mathcal{G}_I^{(r)}(r, \hat{r}, \hat{r}_d) = \frac{1}{2}(r\hat{r} - \hat{r}_d), \tag{116}$$

$$\mathcal{G}_S^{(r)}(\hat{r}, \hat{r}_d) = -\frac{1}{2}(\hat{r} - \hat{r}_d) + \int Dz \log \left( 2 \cosh \left( z\sqrt{r} \right) \right), \tag{117}$$

$$\mathcal{G}_E^{(r)}(r) = \int Dz \log \tilde{H}_{1-r}\left(\sqrt{r}z\right). \tag{118}$$

where we have specialized the entropic term in eq. 93, together with the interaction term and the energetic one in eq. 92 and eq. 94. As a side note, we may observe that in this case the $\hat{r}_d$ order parameter simplifies away with the expression of the interaction term $\mathcal{G}_I$, in both the reference and the tilted systems. This is expected since in the binary case the norm is fixed automatically. We keep this term in the expressions only for uniformity of notation.

### E.2.1 Replica results for the binary perceptron under the uniform distribution

We investigate the output of the replica formalism for the binary case under the uniform distribution, i.e. with $U(s) = 0$. The left panel of Fig. 5 shows the free entropy $\phi_t(q)$ as a function of the overlap $q$ for different values of $t$, with parameters $\alpha = 0.5$ and $\kappa = 0$. The phenomenology observed in this case is completely different from that observed for the spherical case: for all $\alpha$ and all $t$ there is always a second maximum at $q = 1$. Even though it is not always evident from the plots, we showed analytically that that is indeed the case, see Section F. As shown in the figure, the maximum at $q = 1$ eventually becomes the global one for sufficiently large $t$, while there is still a lower-$q$ maximum. As explained in Section 3, this implies that during the sampling procedure the information on the target is not correctly reconstructed, leading to the failure of ASL. The phase diagram shown in the right panel of Fig. 5 shows that this phenomenology is present for all $\alpha > 0$.

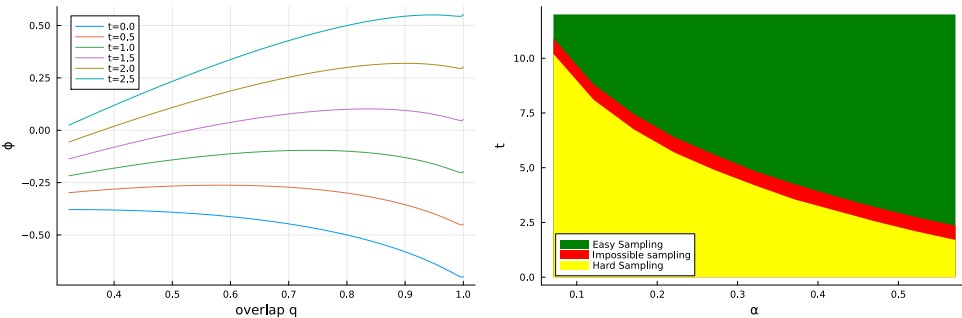

Figure 5: Left: Free entropy $\phi_t(q)$ for the binary perceptron problem for several times. For all $t$, the curves have two local maxima: one at low $q$ that grows with $t$, and a persistent one at $q = 1$. Eventually, the persistent one becomes global, while the lower-$q$ one is still present, which leads to the failure of ASL. Right: Phase diagram of SL sampling for the binary perceptron. At all $\alpha$ the situation is like the one in the left panel: there is a region of $t$ where the lower-$q$ maximum is not the global maximum (red regions) and sampling fails, even though eventually the lower-$q$ maximum eventually disappears (green regions).

### E.2.2 Experimental results for sampling binary perceptron solutions with Log-potential on random data

In Figure 6, we report the probability of sampling a correct configuration from the binary perceptron loss measure with the $\log$-potential, for $T = 0.5$, using the ASL sampling scheme. The asymptotic algorithmic threshold of $\alpha \approx 0.65$ predicted by the RS replica theory (see the left plot in Figure 2) seems consistent with the large $N$ extrapolation of numerical data. Finite size effects are quite strong, though, and we cannot rule out a small gap due to RSB effects.

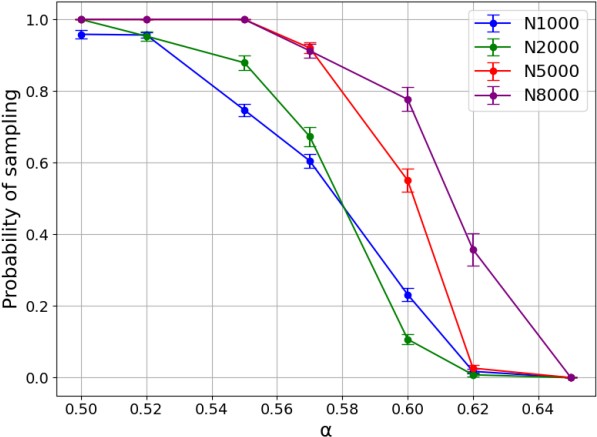

Figure 6: The probability of sampling a solution satisfying the binary perceptron constraints with the ASL algorithm as a function of the constraint density $\alpha$ from a measure with the $\log$-potential with $T = 0.5$. Each point is averaged over 250 simulations.

### E.3 Other plots for $\tau$-annealing

For the $\tau$-annealing scheme described in Section 4.4.2, we show in Figure 7 additional experiments with different values of the number of MC sweeps, eventually also scaling them as $O(N)$. Similar experiments but with the potential $U(s) = -s\Theta(-s)$, and annealing the temperature $T$ instead (from 1 to 0, linearly in the number of sweeps) are presented in Figure 8. It is important to notice that $T$-annealing on $U(s) = -s\Theta(-s)$ fails at large $N$ also when scaling the number of sweeps as $N$, while the $\tau$-annealing reaches an algorithmic threshold of $\alpha \approx 0.55$.

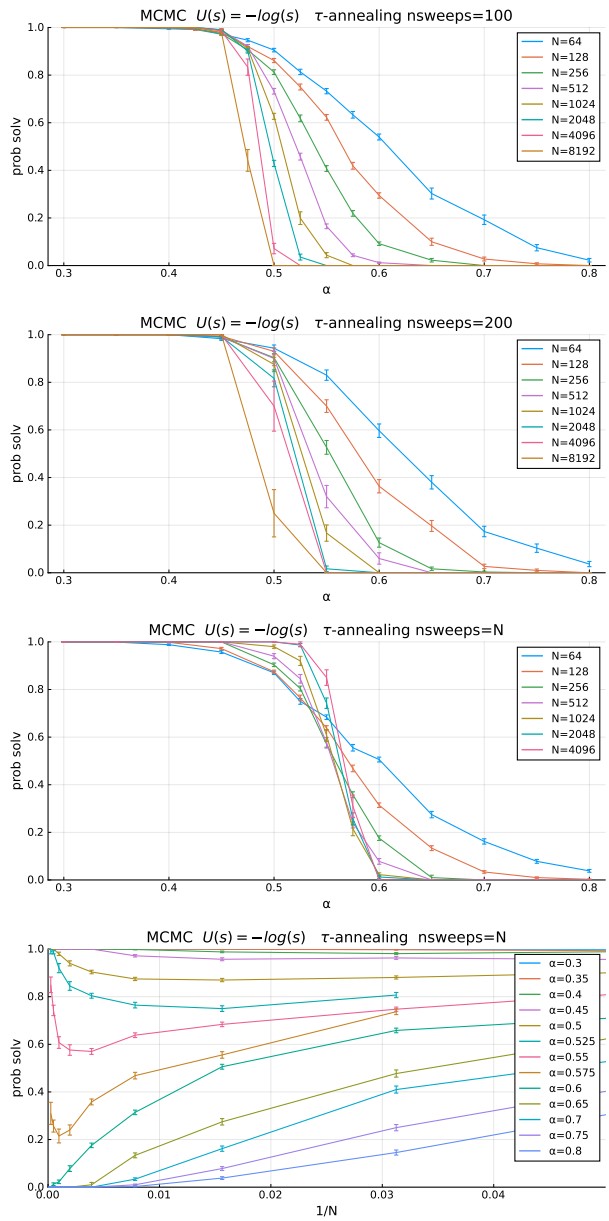

Figure 7: Probability of finding a solution for the $\tau$-annealed MCMC scheme in the Binary Perceptron, with $T = 0.5$.

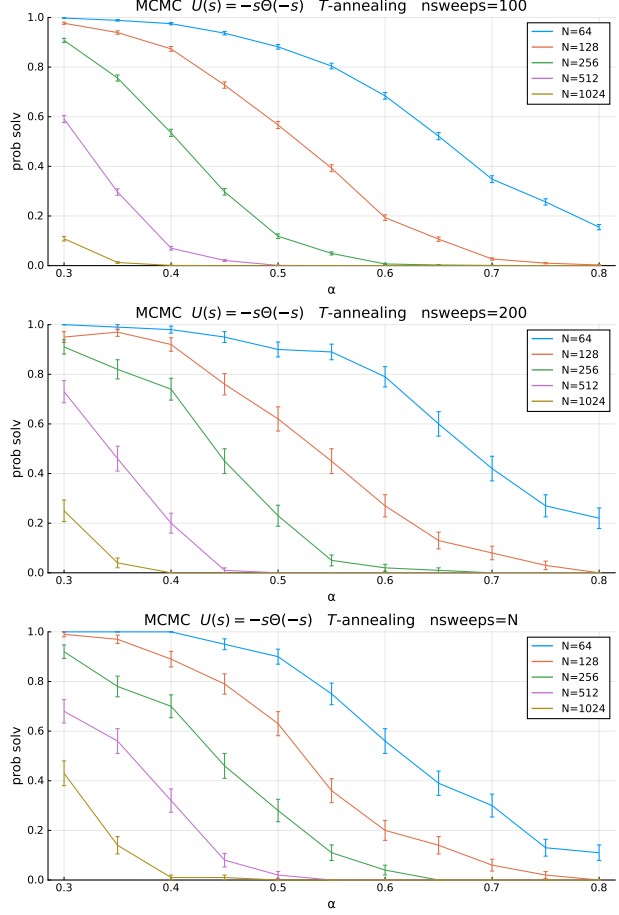

Figure 8: Probability of finding a solution for the $T$-annealed MCMC in the Binary Perceptron with potential $U(s) = -s\Theta(-s)$.

## E.4 Experimental results for sampling binary perceptron solutions with Log-potential on structured datasets

In this section, we use the $\tau$−annealing scheme to perform binary classification on structured datasets. We consider three different datasets: MNIST [Lecun et al., 1998], FashionMNIST [Xiao et al., 2017], and CIFAR10 [Krizhevsky and Hinton, 2009]. For each of them, we choose two classes and assigne to them the labels $\{-1, +1\}$. We then compare $\tau$−annealing scheme with Simulated Annealing on a few losses.

### E.4.1 MNIST dataset

**MNIST patterns classification:** We considered the classes $3$ and $8$. We constructed a binary perceptron architecture with $N = 784$ units (the MNIST image dimension is $28 \times 28$), and we run the $\tau$−annelaing procedure with the $\log$−potential, with a $nsweeps$ in $\{100, 784\}$ and $T = 0.5$. The results have been compared with the $T$−annealing MCMC schemes with potential $U(s) = \Theta(s)$ or $U(s) = -s\Theta(s)$, with the same number of MCMC sweeps. The results are reported in Fig. 9.

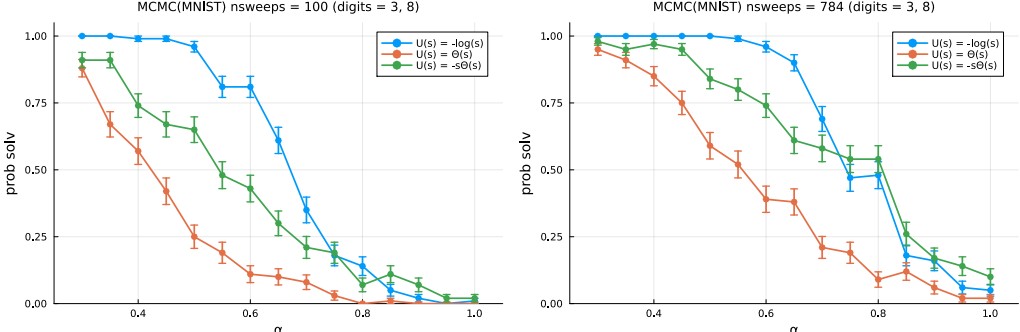

Figure 9: Fraction of correctly classifying MNIST patterns, i.e., the fraction of instances producing zero classification error on digits 3 and 8, as a function of $\alpha = M/N$. The number of different problem instances is fixed to $100$ (the error bars represent the standard deviation estimated over the $100$ instances). The number of MCMC sweeps is fixed to $100$ (left) and $784$ (right).

**Linearly transformed MNIST patterns classification:** The same analysis can be performed in the case in which the patterns that the binary perceptron needs to classify are not directly the MNIST ones, but are obtained by applying a linear transformation to the MNIST images. Given $\{\boldsymbol{x}^{\mu}, y^{\mu}\}_{\mu}$ data in the MNIST dataset, i.e. $\boldsymbol{x}^{\mu} \in \mathbb{R}^{D}$, with $D = 784$, and $y^{\mu} \in \mathbb{R}$, we apply a linear transformation to the patterns $\boldsymbol{x}^{\mu}$ using a projection matrix $F \in \mathbb{R}^{N \times D}$ so that binary perceptron needs to classify $\tilde{\boldsymbol{x}}^{\mu} = F\boldsymbol{x}^{\mu}$. The entries of the projection matrix are $i.i.d.$ sampled from a Normal distribution, i.e. $F_{i,j} \sim \mathcal{N}(0,1) \ \forall i \in \{1, \dots, N\} \ \forall j \in \{1, \dots, D\}$. In this case, the number of MCMC sweeps has been fixed to $N$, and the number of considered instances is 100. In the case of the $\tau-$annealing results with $U(s) = -\log(s)$, the temperature has been fixed to $T = 0.5$. The results are reported in Fig.10 and Fig.11.

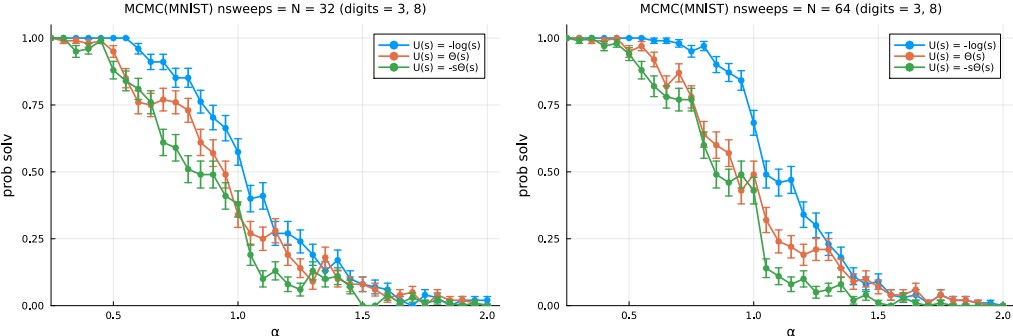

Figure 10: Fraction of correctly classified MNIST patterns (digits = 3 and 8), as a function of $\alpha = M/N$. The number of different problem instances is fixed to $100$, and the number of MCMC sweeps is fixed to $N$ with $N = 32$ (left), and $N = 64$ (right).

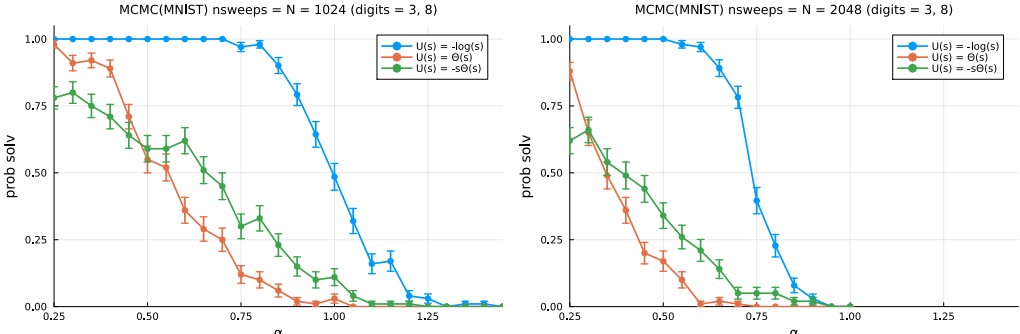

Figure 11: Fraction of correctly classified MNIST patterns (digits 3 and 8), as a function of $\alpha = M/N$. The number of different problem instances is fixed to $100$, and the number of MCMC sweeps is fixed to $N$ with $N = 1024$ (left), and $N = 2048$ (right).

### E.4.2 FashionMNIST dataset

**FashionMNIST patterns classification:** A similar analysis to the previous one has been performed on the FashionMNIST dataset, where we considered classes 5 and 7. We constructed a binary perceptron architecture with $N = 784$ units and ran the $\tau-$annealing procedure with the $\log -$potential, using $nsweeps$ MCMC sweeps in $\{100, 784\}$ and $T = 0.5$. The results have been compared with the $T-$annealing MCMC schemes with potential $U(s) = \Theta(s)$ or $U(s) = -s\Theta(s)$, with the same number of MCMC sweeps. The results are reported in Fig. 12.

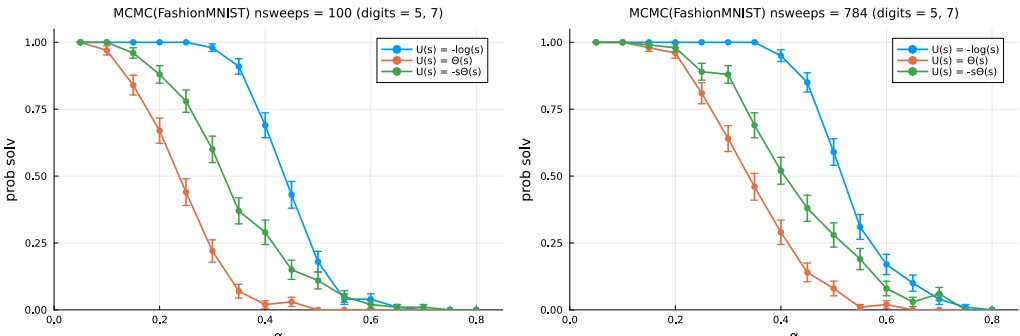

Figure 12: Fraction of correctly classifying FashionMNIST patterns, i.e., fraction of instances producing zero classification error on digits 5 and 7, as a function of $\alpha = M/N$. The number of different problem instances is fixed to $100$ (the error bars represent the standard deviation estimated over the 100 instances). The number of MCMC sweeps is fixed to $100$ (left) and $784$ (right).

**Linearly transformed FashionMNIST patterns classification:** Analogously to the MNIST case, for FashionMNIST, we computed the fraction of correctly solved instances as a function of $\alpha = M/N$, in the case in which we try to classify the linear transformation of the patterns, via the feature matrix $F$, $\{\tilde{\boldsymbol{x}}^\mu\}_{\mu=1,\dots,M}$. As in the case of the MNIST dataset, the number of MCMC sweeps has been fixed to $N$, the temperature for the $\tau-$annealing scheme to $T = 0.5$, and the considered classes are 5 and 7. The results are reported in Fig.13 and Fig.14.

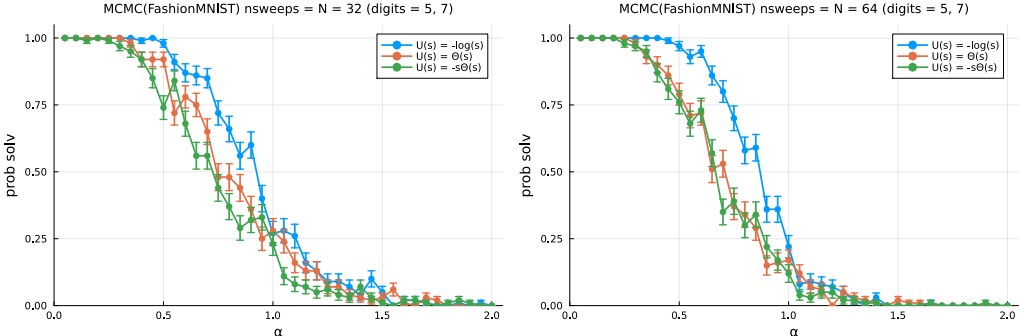

Figure 13: Fraction of correctly classified FashionMNIST patterns (digits 5 and 7), as a function of $\alpha = M/N$. The number of different problem instances is fixed to $100$, and the number of MCMC sweeps is fixed to $N$ with $N = 32$ (left), and $N = 64$ (right).

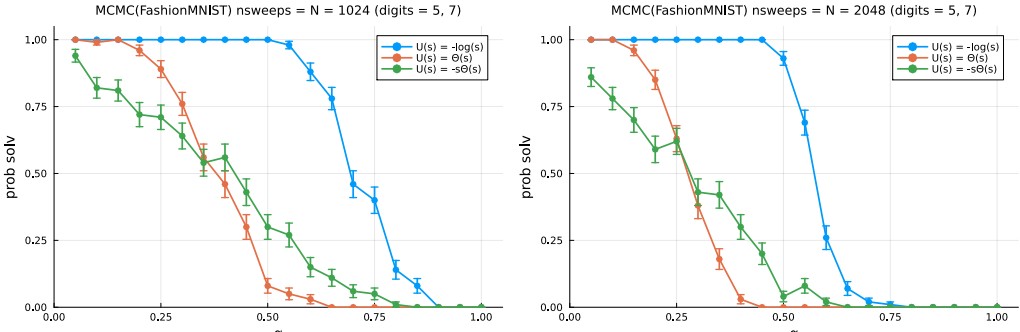

Figure 14: Fraction of correctly classified FashionMNIST patterns (digits 5 and 7), as a function of $\alpha = M/N$. The number of different problem instances is fixed to $100$, and the number of MCMC sweeps is fixed to $N$ with $N = 1024$ (left), and $N = 2048$ (right).

### E.4.3 CIFAR10 dataset

**CIFAR10 patterns classification:** A similar analysis to the one performed for the MNIST and FashionMNIST datasets has been performed over the CIFAR10 dataset, where we considered the classes 3 and 8. We constructed a binary perceptron architecture with $N = 3072$ units (the dimension of the CIFAR10 images is $32 \times 32$ for 3 different color channels), and we ran the $\tau$−annealing procedure with the $\log$−potential, with $nsweeps = \{1000, 3000\}$ MCMC sweeps and $T = 0.5$. The results have been compared with the $T$−annealing MCMC schemes with potential $U(s) = \Theta(s)$ or $U(s) = -s\Theta(s)$, with the same number of MCMC sweeps. The results are reported in Fig. 15.

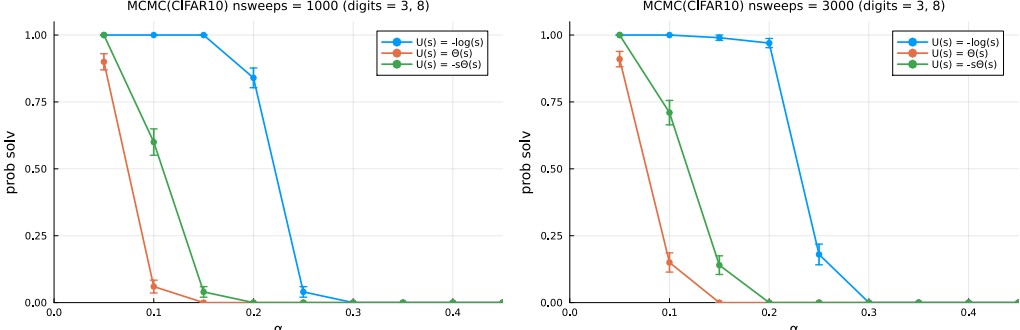

Figure 15: Fraction of correctly classifying CIFAR patterns, i.e., fraction of instances producing zero classification error on digits 3 and 8, as a function of $\alpha = M/N$. The number of different problem instances is fixed to $100$ (the error bars represent the standard deviation estimated over the $100$ instances). The number of MCMC sweeps is fixed to $1000$ (left) and $3000$ (right).

## F   Limiting behavior of the free entropy derivative

In this section, we report the analysis of $\partial_q \phi_t(q)$ as $q \to 1$. We set $q = 1 - \epsilon$ and expand for small $\epsilon$, leading to the results reported in 4.4.

The starting expression is:

$$\phi_t(q, \hat{q}) = -\frac{1}{2}\left(\hat{r}_d + \hat{q}q\right) + \hat{r}r + \frac{1}{2}tq + \mathcal{G}_S(\hat{q}) + \alpha\,\mathcal{G}_E(q) \tag{119}$$

and its derivative is:

$$\frac{\mathrm{d}\phi_t}{\mathrm{d}q} = \frac{t - \hat{q}}{2} + \alpha\frac{\mathrm{d}\mathcal{G}_E(q)}{\mathrm{d}q} \tag{120}$$

We consider the two terms separately.

### F.1   Interaction term

We start from the first (interaction) term. The dependency of $\hat{q}$ on $\epsilon$ can be determined from the saddle point equations. The spherical and binary perceptron models need to be considered separately, since the relevant equation involves $\mathcal{G}_S$, which differs between the two.

#### F.1.1   Spherical Perceptron

In the spherical case, the saddle point equation for $q$ reads:

$$q = 2\frac{\partial\mathcal{G}_S(\hat{q})}{\partial\hat{q}} \tag{121}$$

$$= -\frac{\partial}{\partial\hat{q}}\left(\frac{1}{2}\log(\hat{q} - \hat{r}_d) - \frac{1}{2(\hat{q} - \hat{r}_d)}\left(\hat{q} + 2\hat{r}\frac{(\hat{q} - \hat{r})}{(\hat{r} - \hat{r}_d)} + \frac{(\hat{q} - \hat{r})^2}{(\hat{r} - \hat{r}_d)}\left(\frac{\hat{r}}{\hat{r} - \hat{r}_d} + 1\right)\right)\right), \tag{122}$$

$$= -\frac{1}{\hat{q} - \hat{r}_d} + \frac{2\hat{r} - \hat{r}_d}{(\hat{r} - \hat{r}_d)^2}. \tag{123}$$

We can now use the saddle point for the reference system, see Section B.4.5, to rewrite the expression of $q$, concluding that $\hat{q}$ diverges as

$$\hat{q} = \frac{1}{\epsilon} + \hat{r}_d \tag{124}$$

with $c$ constant. This shows that the part of the derivative due to the interaction term diverges as $O(\epsilon^{-1})$. The sign of the derivative is negative, meaning that the free entropy has a local minimum at $q = 1$ unless this is overcome by the energetic contribution (which will not be the case with $U(s) = 0$, as we will show below).

### F.1.2 Binary perceptron

An analogous computation for the binary perceptron case, already presented in Huang and Kabashima [2014], results in:

$$\frac{1+q}{2} = \frac{\partial}{\partial \hat{q}} \sum_{w^*=\pm 1} \int Dz\, D\gamma\, \frac{e^{\gamma\sqrt{\hat{r}}w^*}}{2\cosh(\gamma\sqrt{\hat{r}})} \log\left(2\cosh\left(z\sqrt{\hat{q}-\hat{r}} + \gamma\sqrt{\hat{r}} + (\hat{q}-\hat{r})\,w^*\right)\right) \tag{125}$$

$$= \sum_{w^*=\pm 1} \int Dz\, D\gamma\, \frac{e^{\gamma\sqrt{\hat{r}}w^*}}{2\cosh(\gamma\sqrt{\hat{r}})} \tanh\left(z\sqrt{\hat{q}-\hat{r}} + \gamma\sqrt{\hat{r}} + (\hat{q}-\hat{r})\,w^*\right)\left(\frac{z}{2\sqrt{\hat{q}-\hat{r}}} + w^*\right) \tag{126}$$

As for the spherical case, $\hat{q} \to \infty$ as $q \to 1$. We can then rewrite the previous expression as

$$1 - \frac{1}{2}\epsilon = 1 - e^{-2(\hat{q}-\hat{r})} \int Dz\, D\gamma\, \frac{e^{\gamma\sqrt{\hat{r}}}e^{-2\left(z\sqrt{\hat{q}-\hat{r}}+\gamma\sqrt{\hat{r}}\right)} - e^{-\gamma\sqrt{\hat{r}}}e^{2\left(z\sqrt{\hat{q}-\hat{r}}+\gamma\sqrt{\hat{r}}\right)}}{\cosh(\gamma\sqrt{\hat{r}})} \tag{127}$$

and thus

$$\hat{q} = -\frac{1}{2}\log\left(\frac{\epsilon}{2}\right) + C \tag{128}$$

$$C = \hat{r} + \frac{1}{2}\log\left(\int Dz\, D\gamma\, \frac{e^{\gamma\sqrt{\hat{r}}}e^{-2\left(z\sqrt{\hat{q}-\hat{r}}+\gamma\sqrt{\hat{r}}\right)} + e^{-\gamma\sqrt{\hat{r}}}e^{2\left(z\sqrt{\hat{q}-\hat{r}}+\gamma\sqrt{\hat{r}}\right)}}{\cosh(\gamma\sqrt{\hat{r}})}\right). \tag{129}$$

This shows that, as $q \to 1$, the part of the derivative that comes from the interaction term also diverges in the binary case, but this time logarithmically, as $O(\log \epsilon)$. The sign of the derivative is again negative, meaning that the free entropy has a local minimum at $q = 1$ unless the effect is overcome by the energetic contribution. As we shall show, the energetic contribution does indeed overcome this effect and produce a local maximum at $q = 1$ unless the potential $U(s)$ diverges at the origin.

### F.2 Energetic term

Next, we compute the derivative of the energetic term. This does not depend on the model. We expand $\mathcal{G}_E(1-\epsilon)$ for small $\epsilon$ and then study $\left(\mathcal{G}_E(1-\epsilon) - \mathcal{G}_E(1)\right)/\epsilon^3$, keeping in mind that:

$$\frac{d\mathcal{G}_E}{dq} = -\frac{d\mathcal{G}_E}{d\epsilon} \tag{130}$$

We perform the expansion, trying to keep the setting general with respect to the potential $U$. Indeed, our only starting assumption is that $U$ is twice differentiable over $(0, \infty)$.

The energetic term, after some manipulations, and using $q = 1 - \epsilon$, can be written as:

$$\mathcal{G}_E = \int D\gamma Dz\, \frac{\mathcal{N}(z,\gamma)}{\mathcal{D}(\gamma)} \tag{131}$$

$$\mathcal{D}(\gamma) = \int_{\frac{\gamma\sqrt{r}}{\sqrt{1-r}}}^{\infty} D\lambda\, e^{-\beta U\left(\lambda\sqrt{1-r}-\gamma\sqrt{r}\right)} \tag{132}$$

$$\mathcal{N}(z,\gamma) = \int_{\frac{\gamma\sqrt{r}}{\sqrt{1-r}}}^{\infty} D\lambda\, e^{-\beta U\left(\lambda\sqrt{1-r}-\gamma\sqrt{r}\right)} \mathcal{A}(z,\gamma,\lambda) \tag{133}$$

$$\mathcal{A}(z,\gamma,\lambda) = \log \int_{-\frac{a(z,\gamma,\lambda,u)}{\sqrt{\epsilon}}}^{\infty} Du\, e^{-\beta U\left(u\sqrt{\epsilon}+a(z,\gamma,\lambda,u)\right)} \tag{134}$$

$$a(z,\gamma,\lambda,u) = -\sqrt{r}\gamma - z\sqrt{1-r-\epsilon}\sqrt{\frac{\epsilon}{1-r}} + \sqrt{1-r}\lambda - \frac{\epsilon\lambda}{\sqrt{1-r}} \tag{135}$$

---

[3]To the sake of simplicity, from now on, we do not make explicit the dependence of $\mathcal{G}_E$.

We first perform two change of variables, in both $\mathcal{N}$ and $\mathcal{D}$, from $\lambda$ to $\rho = -\gamma\sqrt{r} + \lambda\sqrt{1-r}$:

$$\mathcal{D}(\gamma) = \frac{1}{\sqrt{1-r}} \int_0^\infty d\rho\, G\left(\frac{\rho + \gamma\sqrt{r}}{\sqrt{1-r}}\right) e^{-\beta U(\rho)} \tag{136}$$

$$\mathcal{N}(z,\gamma) = \frac{1}{\sqrt{1-r}} \int_0^\infty d\rho\, G\left(\frac{\rho + \gamma\sqrt{r}}{\sqrt{1-r}}\right) e^{-\beta U(\rho)} \tilde{\mathcal{A}}(z,\gamma,\rho) \tag{137}$$

$$\tilde{\mathcal{A}}(z,\gamma,\rho) = \log \int_{-\frac{\tilde{a}(z,\gamma,\rho,u)}{\sqrt{\epsilon}}}^\infty Du\, e^{-\beta U\left(u\sqrt{\epsilon}+\tilde{a}(z,\gamma,\lambda,u)\right)} \tag{138}$$

$$\tilde{a}(z,\gamma,\rho,u) = \rho - z\sqrt{\epsilon}\sqrt{1 - \frac{\epsilon}{1-r}} - \epsilon\frac{\rho + \gamma\sqrt{r}}{\sqrt{1-r}} \tag{139}$$

where $G(x) = \frac{e^{-\frac{x^2}{2}}}{\sqrt{2\pi}}$.

Next, we expand the potential $U$ inside the expression of $\tilde{\mathcal{A}}$ for small $\epsilon$ up to the first order:

$$U\left(u\sqrt{\epsilon} + \tilde{a}(z,\gamma,\lambda,u)\right) \approx U(\rho) + \sqrt{\epsilon}U'(\rho)(u-z) - \epsilon\frac{U'(\rho)}{1-r}\left(\sqrt{r}\gamma + \rho\right) + \frac{1}{2}\epsilon(u-z)^2 U''(\rho) \tag{140}$$

This allows us to compute the integral $\tilde{\mathcal{A}}$, which converges as long as $1 + U''(\rho)\beta\epsilon \geq 0$ (which is obviously always true if $U$ is convex). After some explicit integration and further expansions, we arrive at:

$$\tilde{\mathcal{A}}(z,\gamma,\rho) \approx \mathcal{A}_0 + \sqrt{\epsilon}\mathcal{A}_{1/2} + \epsilon\mathcal{A}_1 + \log H\left(z - \frac{\rho}{\sqrt{\epsilon}}\right) \tag{141}$$

$$\mathcal{A}_0 = -\beta U(\rho) \tag{142}$$

$$\mathcal{A}_{1/2} = \beta z U'(\rho) \tag{143}$$

$$\mathcal{A}_1 = \beta\left(-\frac{U''(\rho)}{2}\left(1 + z^2\right) + \frac{\beta(U'(\rho))^2}{2} + \frac{U'(\rho)\left(\rho + \gamma\sqrt{r}\right)}{1-r}\right) \tag{144}$$

At this point, we can observe that the term $\mathcal{A}_{1/2}$ does not contribute because it gets canceled by the integral over $z$. Indeed, with further manipulations and changes of variables we arrive at:

$$\mathcal{G}_E = \int D\gamma\, \frac{\tilde{\mathcal{N}}(\gamma)}{\tilde{\mathcal{D}}(\gamma)} \tag{145}$$

$$\tilde{\mathcal{D}}(\gamma) = \int_0^\infty d\rho\, G\left(\frac{\rho + \gamma\sqrt{r}}{\sqrt{1-r}}\right) e^{-\beta U(\rho)} \tag{146}$$

$$\tilde{\mathcal{N}}(\gamma) = \int_0^\infty d\rho\, G\left(\frac{\rho + \gamma\sqrt{r}}{\sqrt{1-r}}\right) e^{-\beta U(\rho)}\left[\mathcal{A}_0 + \epsilon\tilde{\mathcal{A}}_1\right] + \tag{147}$$

$$+ \sqrt{\epsilon}\int_0^\infty d\tau\, G\left(\frac{\tau\epsilon + \gamma\sqrt{r}}{\sqrt{1-r}}\right) e^{-\beta U(\tau\epsilon)} \int Dz \log H(z - \tau)$$

$$\mathcal{A}_0 = -\beta U(\rho) \tag{148}$$

$$\tilde{\mathcal{A}}_1 = \beta\left(-U''(\rho) + \frac{\beta(U'(\rho))^2}{2} + \frac{U'(\rho)\left(\rho + \gamma\sqrt{r}\right)}{1-r}\right) \tag{149}$$

We can observe that $\mathcal{G}_E$ receives three contributions from $\tilde{\mathcal{N}}$. The ones from $\mathcal{A}_0$ and $\tilde{\mathcal{A}}_1$ are straightforward: they represent a zero-order and a first-order contributions. In particular the first one gives us the limiting value $\mathcal{G}_E^0 = \lim_{\epsilon\to 0} \mathcal{G}_E$. The first-order term then gives a finite contribution to the derivative.

Thus, the only way to avoid the local maximum at $q = 1$ in the binary perceptron case is to choose a potential $U(s)$ that diverges for $s \to 0$. Then the scaling of the last term with $\epsilon$ can be manipulated.

For the specific choice $U(s) = -\log(s)$ we can easily see that we obtain an additional factor $\epsilon^{\beta/2}$, which makes the term $O\left(\epsilon^{\frac{\beta+1}{2}}\right)$ and the derivative $O\left(\epsilon^{\frac{\beta-1}{2}}\right)$. For $\beta > 1$, this term therefore becomes sub-dominant and it does not contribute to the derivative: we revert to the situation where the dominant contribution no longer comes from the energetic term, but from the interaction term, and it has the right sign to ensure that $q = 1$ will correspond to a local minimum rather than a maximum.

