# OpenReview forum: "Generative diffusion for perceptron problems: statistical physics analysis and efficient algorithms"
_NeurIPS.cc/2025/Conference — NeurIPS 2025 poster_

### Official Review · Reviewer_SCko · 2025-06-26

**Clarity:** 1
**Significance:** 3
**Originality:** 4
**Rating:** 3
**Confidence:** 4

**Summary:**

The paper presents a theoretical extension of the analysis of the Stochastic Localisation framework through replica calculations. In particular, through the study of free entropy, one is able to asymptotically determine thresholds for sampling feasibility via AMP, being the latter conjectured to be optimal for the Bayesian denoising task in a high-dimensional setting. Furthermore, the authors apply their method to the perceptron model, in the spherical and binary weights setup. The spherical case is studied in the case of a uniform distribution on the solution space (non-convex), discussing AMP limits. The binary case is further explored: the authors show that with a peculiar choice of the potential $U(s)$ it is possible to circumvent the issue of a second maximum in the free entropy function, which would make AMP inapplicable. However, they stress that in the presence of structured data AMP would fail either, and it would be necessary to use MCMC; hence, they propose a potential $U_\tau(s)$ that is a sort of regularised version of $-\log s$ in order for MCMC to be applied and anneal $\tau\to 0$, showing such a procedure is more effective than standard $T$ annealing.

**Questions:**

1. The paper’s style sits ambiguously between a physics-oriented and a mathematically rigorous presentation, making it difficult to follow for readers outside the core replica theory community. In this form the paper needs, in my opinion, a major reformatting, in accordance with the Limitations I described above.
2. Key theoretical results lack formal statements (e.g., propositions or theorems) and are not clearly distinguished from prior work such as Ghio et al. (2024). Could the authors explicitly state these results in a formal way to better highlight the novelty and assumptions of their contributions? Even if they prefer a physics-oriented presentation, such highlighting would be beneficial.
3. The connection to the broader machine learning community currently appears somewhat limited, being mostly confined to the introduction and a few remarks on AMP. In particular, the MCMC component—presented as a key contribution—is discussed rather briefly and remains somewhat disconnected from the wider context of modern sampling techniques in ML. It would significantly enhance the paper’s impact if the authors could elaborate on how their results relate to existing methods in the sampling literature, and clarify both the practical relevance and the limitations of their approach in more realistic ML settings. While additional examples are not necessary beyond those already included, a stronger effort to draw connections or analogies with practical ML applications would be greatly appreciated, especially given the acknowledged limitations of AMP and ASL in real-world scenarios.

**Ethical Concerns:**

["NO or VERY MINOR ethics concerns only"]

**Final Justification:**

I appreciate the authors’ detailed and thoughtful response, which carefully addresses my concerns regarding clarity, structure, and theoretical positioning. Their planned revisions—particularly toward a more physics-oriented and accessible presentation—would, in my view, significantly improve the manuscript. However, since I cannot evaluate the revised version directly within the constraints of the NeurIPS review process, I am not in a position to raise my score. I nonetheless believe the proposed changes are promising and encourage the authors to pursue them, as they would strengthen both the clarity and impact of the work.

**Limitations:**

Yes, they recognise the limitations inherited from AMP and also from replica calculations.

**Quality:**

2

**Strengths And Weaknesses:**

**Strenghts**
- The content of the paper is original, and the claims made in the introduction are convincingly supported throughout the manuscript.
- From a theoretical standpoint, the results are sound and well validated, even though they rely on the replica method, which the authors correctly acknowledge as “non-rigorous but extensively validated.”
- The theoretical predictions are empirically verified in a synthetic setup that, while not of direct practical relevance to modern generative models, provides a reasonably novel benchmark for the application of replica methods in machine learning.
- The authors are transparent about the limitations of the proposed theoretical and algorithmic methods, particularly regarding the use of AMP in realistic data scenarios, which is appropriately acknowledged in several parts of the paper.

**Weaknesses**

#### **1. Style and Structure**

The paper sits stylistically between a physics-oriented and a mathematically formal presentation. To improve clarity and accessibility:

- If the authors wish to target the *physics-oriented style*, it would be helpful to enhance the intuition behind the derivations and the results, both theoretical and experimental. This should be feasible, as the treated examples are standard and well-known. This would also align better with a broad conference audience rather than a narrow community of replica theory experts.
- If instead aiming for a more *mathematically rigorous style*, the paper should be revised to adopt a more formal structure, including clearly stated propositions, theorems, assumptions, etc.

Currently, the hybrid format makes the paper difficult to follow — even for readers familiar with replica theory but not embedded in the specialized community. In its present form, the structure is not well-suited to either tradition.

#### **2. Theoretical Contribution**

- The level of generality beyond *Ghio et al. (2024)* is not entirely clear. To highlight the originality of your contribution, it would be very helpful to explicitly state a proposition from *Ghio et al. (2024)* and then present your generalization as a formal result, ideally in a comparative form. This would greatly assist readers in understanding the novelty of your approach.
- The discussion on MCMC is presented as a major contribution in both the abstract and the introduction, and it indeed forms an interesting bridge to the sampling community. However, the space dedicated to this result is extremely limited. I recommend expanding the explanation and clarifying its implications and connections to the rest of the work.

#### **3. Presentation of AMP and Replica Results**

- Section 4.2, which introduces AMP, feels isolated and incomplete. It could either be moved to the appendix or expanded to give a more self-contained and pedagogical explanation, especially clarifying the role of $\phi_{{in}}$ and $\phi_{{out}}$, which are introduced but never used later.
- A similar issue arises with the replica computations (e.g., Equations 15–18): they occupy a significant amount of space, but are not discussed or interpreted in the main text. Either move them to the appendix or incorporate them into the narrative with interpretation and relevance.

---

> ### Author Rebuttal · Authors · 2025-07-30
>
> We thank the reviewer for their comments. We appreciate that the reviewer values the significance and novelty of our work, while their main concerns are mostly about style and clarity.
>
> We agree that statistical‐physics methods used in machine learning may be unfamiliar to some (most!) readers; therefore, we will strive for a gentler introduction to techniques and the concepts employed.
>
> Unfortunately, we don’t have the opportunity to produce an updated version of the manuscript during the rebuttal process, but we will take into consideration the reviewer’s recommendations and improve the readability of the manuscript in the camera-ready version.
>
> In what follows, we answer their main concerns in detail. We hope that in the light of these answers, the reviewer will be willing to upgrade the overall score.
>
> **Weaknesses and Questions**
>
> > **1. Style and Structure**
> >
> >
> > The paper sits stylistically between a physics-oriented and a mathematically formal presentation. To improve clarity and accessibility...
> >
>
> and also
>
> > The paper’s style sits ambiguously between a physics-oriented and a mathematically rigorous presentation, …
> >
>
> We agree that the clarity can (and will) be improved. We will embrace the physics-oriented style, which is also quite aligned with certain branches of the theoretical ML literature, and make every effort to enhance the intuition behind the derivations and the results, as suggested by the reviewer, in order to make it more accessible to a broader audience.
>
> > **2. Theoretical Contribution**
> >
> > - The level of generality beyond *Ghio et al. (2024)* is not entirely clear…
>
> and also
>
> > Key theoretical results lack formal statements (e.g., propositions or theorems) and are not clearly distinguished from prior work such as Ghio et al. (2024)…
> >
>
> We thank the reviewer for these well-motivated comments. The work of Ghio et al. is indeed very close and provides inspiration to our work. In the revised version of the paper, we will highlight more formally the differences between our work and Ghio et al., as we now discuss.
>
> Let us first emphasise the **main limitation of Ghio et al.** that concerns us:
>
> - Their formalism is limited to planted ensembles, where the reference configuration (the one eventually sampled at the end of the generation process) can always be assumed to be the planted one in the theoretical analysis, since it is statistically identical to any other sample from the target distribution thanks to the Bayesian structure of the problem. This allows to avoid dealing with hard-to-compute normalizing factor Z in the denominator.
> - The authors also consider some non-planted settings. However, their analysis hinges on the fact that they are contiguous to their planted versions (a property known as quiet planting). This is not a general feature but is limited to specific problems and regions of the parameter space. These details are thoroughly explained in Section 1 of the Supplementary Information of Ghio et al. Therefore, even in the non-planted settings, they ultimately analyze planted models, which is what their formalism is designed to handle.
>
> **We address this limitation as follows:**
>
> - We introduce a formalism that allows us to deal with generic target distributions, possibly with hard-to-compute normalization factors. It is based on a double application of the replica trick. This greatly expands the range of applicability of the approach.
> - Thanks to the Bayesian structure of the generative diffusion process, we show that the Nishimori conditions hold and simplify the solution of the saddle point equations.
>
> Moreover, **we provide the following further contributions:**
>
> - In the case of the binary weight perceptron, we don’t just confirm the difficulty of sampling from the uniform distribution: our analysis lets us pinpoint precisely the failure mode of the ASL procedure, and identify a minimal correction that overcomes this problem. While some heuristic solvers for the binary perceptron are known (e.g. Braunstein Zecchina [2006], Baldassi Braunstein [2015], and others), this is the first approach that produces samples from a well-defined and theoretically motivated target distribution over binary perceptron solutions.
> - On the algorithmic side, the failure mode analysis also inspired a practical MCMC annealing scheme as a robust solver for generic (non-random) instances of the problem, which is far more effective than standard simulated annealing.
>
> **[Braunstein Zecchina 2006]** *Learning by message passing in networks of discrete synapses.* *PRL*
>
> **[Baldassi Braunstein 2015]** *A max-sum algorithm for training discrete neural networks*. *JSTAT*
>
> **[Ghio et al 2024]** Ghio, Dandi, Krzakala, and Zdeborová, *Sampling with flows, diffusion, and autoregressive neural networks from a spin-glass perspective*, *PNAS*
>
> > The discussion on MCMC is presented as a major contribution in both the abstract and the introduction… However, the space dedicated to this result is extremely limited…
> >
>
> Unfortunately we had to limit that section due to space constraints, and provided a few additional experiments in Appendix C.3. In the revised version, we will comment on possible extensions of our MCMC scheme to other constraint satisfaction problems, discuss the geometry of the solution space on which the tilted measure focuses (dense cluster of solutions) and provide experiments on structured problems.
>
> > **3. Presentation of AMP and Replica Results**
> >
> > - Section 4.2, which introduces AMP, feels isolated and incomplete. It could either be moved to the appendix or …
> > - A similar issue arises with the replica computations (e.g., Equations 15–18)…
>
> We agree with the referee and we will move to appendices the definitions of the channel free entropies, making room for a more pedagogical explanation of AMP.
>
> On the other hand, we report the free entropy given by the replica computation in the main text mainly to demonstrate how the analysis of the high-dimensional system is reduced to the analysis of a simple low-dimensional function. We will provide more intuition around the different terms appearing in the formulas.
>
> > The connection to the broader machine learning community currently appears somewhat limited... In particular, the MCMC component—presented as a key contribution—is discussed rather briefly and remains somewhat disconnected from the wider context of modern sampling techniques in ML..
> >
>
> We thank the reviewer for this thoughtful comment. We agree that clarifying the broader ML relevance of our MCMC-based approach would enhance the paper. In the final version, to the extent allowed by space constraints, we plan to touch on the following aspects:
>
> 1. **Connection to Modern ML Sampling:**
>
>     In the discussion, we will more explicitly connect our work to modern efforts in score-based and diffusion-based generative modeling, and specifically to recent hybrid samplers that combine score estimation with MCMC techniques (e.g., Grenioux et al. [2024]; Vargas et al. [2023]).
>
> 2. **Practical Relevance and Limitations:**
>
>     We will clarify that ASL, although limited in scope, can be viewed as a diagnostic and conceptual tool for understanding when and how diffusion methods can succeed in sampling hard posterior-like distributions. Its analysis can suggest how to adjust the scheme when it fails, as well as produce algorithmic fallouts such as $\tau$-annealing MCMC.
>
> 3. **Broader ML Implications:**
>
>     The models that we investigate in great detail are non-convex neural networks that are of particular interest to the ML community and are also prototypical examples of CSPs. For CSPs, our approach could be extended to other settings and produce tilting strategies that allow for efficient samplers and solvers.
>
>     In ML contexts, dense regions of solutions / wide minima of the training loss landscape (that are targeted by the introduced tilted measure) have been linked to improved generalization properties (Chaudari et al [2019], Foret et al [2021]). It could be the case that the sampling schemes developed for binary perceptrons can be applied to quantized neural networks: $\tau$-annealing could potentially improve training and distillation procedures for quantized neural networks.
>
>
> We hope these remarks will help clarify the relevance of our work to the broader ML community and demonstrate the practical implications and novelty of the proposed MCMC approach.
>
> **[Grenioux et al 2024]** Grenioux, Noble, Gabrié, and Durmus, *Stochastic localization via iterative posterior sampling*, 2024.
>
> **[Vargas et al 2023]** Vargas, Grathwohl, and Doucet, *Denoising diffusion samplers,* 2023.
>
> **[Chaudari et al 2019]** Chaudhari, Choromanska, Soatto, LeCun, Baldassi,
> Borgs, Chayes, Sagun, and Zecchina, *Entropy-sgd: Biasing gradient descent into wide valleys*. *JSTAT*
>
> **[Foret et al. 2021]** Forêt, Kleiner, Mobahi, Neyshabur, *Sharpness‐Aware Minimization for Efficiently Improving Generalization,* ICLR

---

> > ### Comment · Reviewer_SCko · 2025-08-01
> >
> > I would like to thank the authors for their thoughtful and detailed response. I truly appreciate the care with which they addressed my comments, particularly those concerning the style and structure of the manuscript. Their intention to adopt a clearer, more physics-oriented presentation—while reinforcing intuition and accessibility—is, in my view, a very welcome direction that would significantly improve the readability and reach of the paper.
> > The clarifications regarding the theoretical contributions, especially in relation to Ghio et al. (2024), are well argued and demonstrate a meaningful step forward. I also found the discussion around the ASL method and the proposed MCMC scheme compelling, as it highlights both conceptual insights and promising algorithmic developments.
> >
> > However, I believe the revisions needed to fully address these issues are substantial and would effectively result in a significantly reworked manuscript. Unfortunately, given the structure of the NeurIPS review process, I do not have access to an updated version to evaluate these changes directly—something I completely understand is beyond the authors’ control.
> > For this reason, while I appreciate the authors’ intentions and believe the revised version would be much improved, I do not feel in a position to raise my score. That said, I am confident that by implementing the proposed changes, the work could be very well received in this or a similar venue and have a broader impact. I encourage the authors to pursue this direction, as it would benefit both the clarity of the paper and its visibility within the community.

---

### Official Review · Reviewer_5GDK · 2025-06-26

**Clarity:** 2
**Significance:** 1
**Originality:** 2
**Rating:** 3
**Confidence:** 1

**Summary:**

This paper presents a theoretical framework to analyze the efficiency of sampling the solution space of high-dimensional random perceptron problems using generative diffusion algorithms. The authors employ a statistical physics approach, using the non-rigorous but well-validated replica method, to predict the performance of Algorithmic Stochastic Localization (ASL) , where Approximate Message Passing (AMP) serves as the score approximator.

**Questions:**

* Could you elaborate on the motivation for studying the spherical non-convex perceptron? While it is described as a model for "glassy behavior", what specific algorithmic or theoretical challenges does it present that makes it a crucial case study for your generative diffusion framework?
* The τ-annealing MCMC scheme is proposed as a robust alternative to AMP for the binary perceptron. Could you provide a more detailed analysis of its scalability, perhaps by discussing its computational complexity and how its performance (e.g., time-to-solution) compares to other state-of-the-art heuristic solvers for this problem, especially as the system size N increases?

**Ethical Concerns:**

["NO or VERY MINOR ethics concerns only"]

**Final Justification:**

I have read the rebuttal and decided to lower my confidence score based on my unfamiliarity with the subject.

**Limitations:**

yes.

**Paper Formatting Concerns:**

no concern.

**Quality:**

3

**Strengths And Weaknesses:**

## Strength

* The problem consists of novel formulation of perceptron problems. A key strength is the generalization of prior work (e.g., Ghio et al. [2024]) to handle unnormalized target densities.
* The authors provide a detailed asymptotic analysis for both the spherical and binary perceptron models. The main contributions are explicitly stated and well-supported by the derivations.

## Weaknesses

* **Significance.** The method heavily relies on approximate message passing, which is a well-established concept but also has limited use in real-world application. The authors claimed that the method provides a fast enough algorithm for their "synthetic" setting (pp. 4). Therefore, I have a concern about its significance since I have the impression from literature that the solution acquisition method for perceptrons will not scale well. Since it relies on the quality of the sample of MCMC method, it is likely that the scalability of the proposed method is severely limited.
* **Novelty.** Most of the concepts outlined in this paper are well-known and this paper does not produce new distinct theorems. Therefore, I believe the contributions of this paper are more in the empirical side, and I think a new formalism is in Eqs. (15-17) serve that purpose. However, the authors do not clearly articulate the reason why we look for Spherical Non-Convex Perceptrons. The novelty of this paper is not clearly expressed and the motivation of the work needs a comprehensive revision.
* **Experiment.** The paper needs some experimental results on scalability, and the application side of spherical nonconvex perceptions. I also suggest the authors to include demonstrative illustrations and figures in Section 1 or 3, which help understanding of broad audience

---

> ### Author Rebuttal · Authors · 2025-07-30
>
> We thank the reviewer for their comments. We try to answer in detail to the concerns they raised. Some of the questions are not entirely clear to us; therefore, we provide additional context to try to disambiguate the conversation. We hope our responses adequately address the raised concerns, and we would be grateful if the reviewer could consider revising their score considering these clarifications.
>
> > **Significance.** The method heavily relies on approximate message passing, which is a well-established concept but also has limited use in real-world application. The authors claimed that the method provides a fast enough algorithm for their "synthetic" setting (pp. 4). Therefore, I have a concern about its significance since I have the impression from literature that the solution acquisition method for perceptrons will not scale well. Since it relies on the quality of the sample of MCMC method, it is likely that the scalability of the proposed method is severely limited.
>
> Unfortunately, here we are not sure we fully understand the referee’s comment. There may have been a misunderstanding regarding the setting and goals of our work, which we try to clarify below.
>
> In our setting, the target distribution is known up to a normalization factor that is hard to compute. We provide a formalism to assess the limitations of sampling from the target distribution using generative diffusion. The score function is implemented algorithmically using AMP, which is optimal for i.i.d. samples in the large size limit. This setting contrasts with the typical ML setting, where the target distribution is unknown but samples from it are available, and a neural network is trained to approximate the score function based on those samples.
>
> Therefore, we believe that concerns about scalability due to limited sample availability (if this is what the question refers to) do not apply in our setting, as no sampling from the target distribution is required.
>
> > **Novelty.** Most of the concepts outlined in this paper are well-known and this paper does not produce new distinct theorems. Therefore, I believe the contributions of this paper are more in the empirical side, and I think a new formalism is in Eqs. (15-17) serve that purpose. However, the authors do not clearly articulate the reason why we look for Spherical Non-Convex Perceptrons. The novelty of this paper is not clearly expressed and the motivation of the work needs a comprehensive revision.
>
> We don’t provide theorems, we provide precise conjectures with a well-established and exact (although non-rigorous) method, replica theory. The resulting findings are, to the best of our knowledge, novel and not previously reported. We introduce a new general theoretical framework to investigate the ultimate limits of samplability via generative diffusion. As concrete applications, we explore two paradigmatic examples of neural networks and constraint satisfaction problems. We discuss the motivation for investigating the spherical non-convex perceptron model below, in a response to a question.
>
> For further details on the novelty of our work, please see our rebuttal to Referee 6gzG.
>
> > **Experiment.** The paper needs some experimental results on scalability, and the application side of spherical nonconvex perceptions. I also suggest the authors to include demonstrative illustrations and figures in Section 1 or 3, which help understanding of broad audience
>
> We are not entirely sure what the reviewer means by scalability. If it refers to larger system sizes, we believe that the sizes we have already explored are sufficiently large, as our experimental results align perfectly with the theoretical predictions in the infinite-size limit (as verified in the experiments for the negative spherical perceptron setting).
>
> Although we doubt that we will be able to add new figures due to space constraints, in the revised version, we will try to provide a gentler introduction for the broader audience.
>
> > Could you elaborate on the motivation for studying the spherical non-convex perceptron? While it is described as a model for "glassy behavior", what specific algorithmic or theoretical challenges does it present that makes it a crucial case study for your generative diffusion framework?
>
> The simplest model of a neural network, the perceptron as originally defined, is convex. Therefore, the geometry of its solution space is not sufficiently rich to exhibit the features that characterize more complicated, non-convex neural networks. Characterizing analytically the typical behavior of multi-layer neural networks, on the other hand, is very challenging, even when using powerful frameworks such as replica theory. An approach that has been demonstrated to be fruitful is that of considering modified, non-convex perceptron models. Two simple ways to disrupt the convexity of the model are to use a negative value for the stability (negative spherical perceptron) and to constrain the weights to binary values, which are the two models we study in our paper. Both models have been used in the study of the geometry of solution spaces, the effects of overparameterization, and aspects of generalization. Recent work on the negative spherical perceptron has uncovered a star-shaped connectivity of solution clusters [Annesi et al. 2023], a property later observed empirically in deep neural networks trained on MNIST and CIFAR benchmarks [Lin et al. 2024]. Furthermore, Montanari et al. [2024] investigated the generalization behavior of simple non-convex perceptron networks, bridging theoretical predictions and empirical performance in more complex architectures.
>
> **[Annesi et al. 2023]** The star-shaped space of solutions of the spherical negative perceptron, B. Annesi, C. Lauditi, C. Lucibello, E. Malatesta, G. Perugini, F. Pittorino, and L. Saglietti, Phys. Rev. Lett.  2023.
>
> **[Lin et al. 2024]** Exploring Neural Network Landscapes: Star-Shaped and Geodesic Connectivity, Z. Lin, P. Li and L. Wu,  ArXiv (2024).
>
> **[Montanari et al. 2024]** Tractability from overparametrization: The example of the negative perceptron, A. Montanari, Y. Zhong and K. Zhou, *Probab. Theory Relat. Fields*, (2024).
>
> > The τ-annealing MCMC scheme is proposed as a robust alternative to AMP for the binary perceptron. Could you provide a more detailed analysis of its scalability, perhaps by discussing its computational complexity and how its performance (e.g., time-to-solution) compares to other state-of-the-art heuristic solvers for this problem, especially as the system size N increases?
>
> Our experiments already show a finite algorithmic threshold in alpha with $O(1)$ sweeps (which implies $O(N^2)$ complexity), see Fig. 3 (right). Taking $O(N)$ sweeps or more appears to further increase the algorithmic threshold, as shown in Fig. 6.
>
> Other known algorithms that find solutions to binary perceptron problems [Braunstein and Zecchina 2006; Abbe 2022] have complexity at least $O(N^2)$. We stress that ours is the first solver that outputs solutions that are both diverse and under control, as they are sampled from a well-defined target distribution.
>
> **[Braunstein and Zecchina 2006]** Braunstein, A. and Zecchina, R., Learning by message passing in networks of discrete synapses. *Physical Review Letters*, 2006.
>
> **[Abbe 2022]** Abbe, Li, Sly, *Binary perceptron: efficient algorithms can find solutions in
> a rare well-connected cluster*, STOC 2022

---

> ### Comment · Reviewer_5GDK · 2025-08-04
>
> I have read the rebuttal and decided to lower my confidence score based on my unfamiliarity with the subject. I will discuss this with the other reviewers later to determine the final recommendation score.

---

### Official Review · Reviewer_6gzG · 2025-06-27

**Clarity:** 3
**Significance:** 3
**Originality:** 3
**Rating:** 5
**Confidence:** 4

**Summary:**

The paper considers the problem of sampling from a high-dimensional distribution
$$p(\mathbf{w}) = \frac{\psi(\mathbf{w})}{Z},$$
in a setting where only the unnormalized density $\psi(\mathbf{w})$ is known.

The authors study an approach based on generative diffusion (or stochastic localization),
which allows to sample from $p$ by solving a stochastic differential equation (SDE).
At each time $t$, one has to compute the Bayes-optimal estimator of $\mathbf{w}^\star$
from the observation
of $\mathbf{y} = t \mathbf{w}^\star + \sqrt{t} \mathbf{g}$, where $\mathbf{w}^\star \sim p$ and $\mathbf{g} \sim \mathcal{N}(0,1)$.
Based on existing literature, a natural approach to compute this BO estimator is via the Approximate Message-Passing (AMP) algorithm.
Using non-rigorous methods from statistical physics (namely the replica method), the authors show that whether this sampling procedure (called ASL -- Algorithmic Stochastic Localization) succeeds can be characterized
from the study of a one-dimensional potential $\phi_t(q)$, for which they provide an explicit form.

They apply this framework to perceptron-type problems:
focusing on the region where the problem is solvable, the goal is to sample uniformly from the set of solutions.
For the negative spherical perceptron, the analysis of the paper shows that ASL succeeds in (almost) all the replica-symmetric region of the phase diagram.
In the binary perceptron with zero margin, the authors show that instead ASL fails in the whole phase diagram, in agreement with previous works that argue computational hardness of sampling in the binary perceptron.
On the other hand, by tilting the measure over the space of solutions by a well-chosen potential, ASL is shown to succeed at sampling this modified measure in a large region of the phase diagram.
Finally, sampling of this tilted measure is also shown numerically to be achievable by a MCMC algorithm.

**Questions:**

- (Line 36): The motivation of sampling from such unnormalized probability distributions is quite brief, and
  could be developed further.

- (Line 59): I think that [Ghio\&al '24] also considers sampling from non-planted distributions, such as the p-spin model and the solutions of random graph coloring.
  Could the authors correct me, or clarify what they mean?
  On line 139, it is also implied that the specificity of [Ghio\&al '24] is to consider normalized densities, which makes it a bit confusing.
  More generally, I would be potentially willing to change my rating and confidence score
  if the authors were to convince me better regarding the technical novelty of their work with respect to [Ghio\&al '24].

- I think that at several points (e.g. regarding the introduction of the free entropy line 150, or the replica results line 177),
  reading the paper might be complex for an audience which is not expert in statistical physics techniques. It would be good to give a more pedagogical presentation
  of these tools for this audience, which is the vast majority of the NeurIPS attendees.

- In the conclusion, the authors mention that the $\tau$-annealing procedure proposed overcomes ASL's limitation on structured instances of the problem, but this is not explored in the paper.
  I imagine this is rather an informal conjecture made by the authors, by analogy with similar statistical learning problems?
  Moreover, the authors mention that the ideas developed in this paper could enhance "sampling and solving" in other hard CSPs: I wonder how would the present analysis help in solving (i.e. finding a solution to the problem in a given instance)?

Finally, I list here more minor points:

- (Line 66) I believe the negative spherical perceptron was introduced before the work of Franz \& Parisi in 2016.
  While I don't know the earliest references, it is studied for instance in the work "Negative spherical perceptron" (Stojnic '13),
  and is mentioned in Chapter 8 of the book "Mean-field models for spin glasses" (Talagrand '10), see Chapter 8 (for instance in Problem 8.4.3).

- (Line 122) It would be good to mention again in this paragraph the works that introduced ASL, and that are cited in the introduction.

- (Line 128) I would say that in terms of limitations of AMP, time complexity should also be mentioned, it's possibly a stronger limitation in practice than low-dimensionality of the data.

- Equation (12): the introduction of both $T$ and $U$ is a bit redundant, and makes part of the exposition slightly confusing (since $T$ is not really the temperature of the problem, as we are still trying to sample zero-energy states and not the Gibbs measure at temperature $T$). Maybe a different name for $T$ would be best.

- (Line 243) It is quite surprising at first sight that the paratmeters of the reference system are independent of $t, q$ and $\hat{q}$, since they interact via the replica $w_1^\star$: could the authors comment on that?
  Does this suggest some effective decoupling of the term $Z^{-1}$ with the other terms in the replica computation?

- (Line 263) Since the computation here is done assuming that $p$ is RS, wouldn't the whole approach break down beyond the dAT line?
  As such, can the authors conclude
  from this calculation that sampling being hard in a RSB phase follows from their analysis?

- Finally, I found the following typos/misprints:
  - Line 110: I think $p_{\mathbf{h}, t}(\mathbf{w})$ is not really a convolution with the Gaussian noise (this would rather be the distribution of $\mathbf{h}_t$).
  - (Line 233) The wording "quite fast" reads off as a bit vague.
  - (Line 244): "system free entropy"
  - (Line 246): "allow us"
  - Caption of Fig.2: I would say the agreement "suggests" rather than "shows" (although
    this analysis is indeed a convincing argument for the fairness of produced samples).
  - (Line 285) It should be "El Alaoui" rather than "Alaoui"
  - (Line 325) "for SA"
  - (Line 327) "we sample" might be a better word than "we solve"

**Ethical Concerns:**

["NO or VERY MINOR ethics concerns only"]

**Final Justification:**

The authors clarified many of my concerns, especially regarding the novelty with [Ghio & al], see our discussion below.

**Limitations:**

yes

**Paper Formatting Concerns:**

No concern.

**Quality:**

3

**Strengths And Weaknesses:**

Let me clarify that I read through the main text in detail (following the computations and arguments),
but did not check the supplementary material for reasons of time.
Moreover, while I am quite familiar with the statistical physics approaches to learning theory, this is slightly less the case for generative diffusion.

**Strengths --**

- The paper is written in a clear fashion. Concepts are exposed very clearly while
  remaining concise (Section 2 is a good example), and I want to thank the authors for this effort in the writing.
  The technical tools used here (the replica method and the analysis of AMP algorithms) are well presented and detailed.
  The results are well described, e.g. via clear figures.
- The replica computation in this generic setting, as well as the phenomenology of sampling in perceptron models
  are contributions which would be of interest to the audience engaged in similar topics, as this a subject of strong recent activity and the paper
  shows interesting results in this regard (e.g. that sampling a weighted measure over the space of solutions rather than the uniform one can make the problem drastically easier
  in binary perceptrons).


**Weaknesses --**

- The manuscript (both by the subject of study and the presentation) ends up being targeted towards an audience already quite knowledgeable in the statistical physics approach to the theory of statistical learning,
  and because of this the NeurIPS audience might not be the best fit for it in my opinion.
  I tend to think it would be better suited e.g. in venues closer to statistical physics interests.
  On the other hand, I also understand the upsides of introducing part of this audience to these powerful techniques, so I don't consider this a very weak point.
- The main weakness of the paper, in my eyes, is the seemingly small novelty with respect to the analysis of
  [Ghio \& al '24], which conducts a very similar analysis, although the authors claim that it applies only to planted models (line 59) or normalized densities (line 139).
  The replica computation seems very similar in nature, modulo the computation of $Z^{-1}$ with an additional use of the replica method.
  At a high level, this work seems like a fairly close computation, applied to $p$ being the solutions of perceptron problems, rather than e.g. graph coloring or p-spin models which are analyzed in [Ghio\&al '24].

---

> ### Author Rebuttal · Authors · 2025-07-30
>
> We thank the reviewer for raising several important points, which give us the opportunity to highlight the key novelties and strengths of our work. Below, we address each major concern in detail (within the constraints of the rebuttal format) and acknowledge the remaining minor points.
>
> We would be happy to further discuss any remaining concerns, and we would appreciate if the reviewer could consider raising their score in case our responses address their questions adequately.
>
> > The manuscript (both by the subject of study and the presentation) ends up being targeted towards an audience already quite knowledgeable in the statistical physics … On the other hand, I also understand the upsides of introducing part of this audience to these powerful techniques, so I don't consider this a very weak point.
>
> We believe that, given our work involves fundamental neural network models, generative diffusion, sampling problems, and message passing algorithms, a venue such as NeurIPS would allow us to reach a broader audience of potentially interested readers compared to a stat. physics journal.
>
> We acknowledge that our paper does not provide a pedagogical introduction to the statistical physics approach to statistical learning, and it assumes some prior background. Fortunately, thanks to the growing interest of the statistical physics community in machine learning and the openness of the ML community, this knowledge has been spreading in recent years. We hope to contribute to this trend, at least by offering motivation to explore these techniques more deeply.
>
> > The main weakness of the paper, in my eyes, is the seemingly small novelty with respect to the analysis of [Ghio & al '24], which conducts a very similar analysis, although the authors claim that it applies only to planted models (line 59) or normalized densities (line 139) …
>
> and also
>
> > (Line 59): I think that [Ghio & al '24] also considers sampling from non-planted distributions, such as the p-spin model and the solutions of random graph coloring. Could the authors correct me, or clarify what they mean? On line 139, it is also implied that the specificity of [Ghio & al '24] is to consider normalized densities, which makes it a bit confusing. **More generally, I would be potentially willing to change my rating and confidence score if the authors were to convince me better regarding the technical novelty of their work with respect to [Ghio & al '24]**.
>
> We thank the reviewer for these well-motivated comments. The work Ghio et al is indeed very close and provided inspiration to our work.
>
> Let us first highlight in more detail the **main limitation of Ghio et al**  that concerns us:
>
> - Their formalism is limited to planted ensembles, where the reference configuration (the one eventually sampled at the end of the generation process) can always be assumed to be the planted one in the theoretical analysis, since it is statistically identical to any other sample from the target distribution thanks to the Bayesian structure of the problem. This allows to avoid dealing with hard-to-compute normalizing factors Z in the denominator.
> - The reviewer is correct in noting that Ghio *et al.* also analyze the $p$-spin and NAE-SAT models in the non-planted setting. However, their analysis relies on a property of these problems, namely, that they are contiguous to their planted counterparts. This is not a general property: it holds only for specific problems and within certain regions of the parameter space. This phenomenon is known as *quiet planting*, and is explained in detail in Section 1 of the Supplementary Information of Ghio et al. Thus, even in these non-planted settings, they are ultimately analyzing planted models, which is what their formalism is equipped to handle.
>
> **We address this limitation as follows**
>
> - We introduce a formalism that allows us to handle generic target distributions, possibly with hard-to-compute normalization factors. It is based on a double application of the replica trick, which significantly broadens the scope of its applicability.
> - Thanks to the Bayesian structure of the generative diffusion process, we can show that the Nishimori conditions hold and simplify the solution of the saddle point equations.
>
> Moreover, **we provide the following further contributions:**
>
> - In the case of the binary weight perceptron, we don’t just confirm the difficulty of sampling from the uniform distribution: our analysis lets us pinpoint precisely the failure mode of the ASL procedure, and identify a minimal correction that overcomes this problem. While some heuristic solvers for the binary perceptron are known (e.g. Braunstein Zecchina [2006], Baldassi Braunstein [2015], and others), this is the first approach that produces samples from a well-defined and theoretically motivated target distribution over binary perceptron solutions.
> - On the algorithmic side, the failure mode analysis also inspired a practical MCMC annealing scheme as a robust solver for generic (non-random) instances of the problem, which is far more effective than standard simulated annealing.
> - Finally, the models that we investigate are non-convex neural networks that are of particular interest to the machine learning community and are also prototypical examples of constraint satisfaction problems. Dense regions of solutions / wide minima of the training loss landscape have been linked to improved generalization properties (Chaudari et al [2019], Foret et al [2021]). We thus believe that our theoretical and algorithmic insights go beyond the simple perceptron models we consider.
>
> **[Braunstein Zecchina 2006]** *Learning by message passing in networks of discrete synapses.* *PRL*
>
> **[Baldassi Braunstein 2015]** *A max-sum algorithm for training discrete neural networks*. *JSTAT*
>
> **[Ghio et al 2024]** Ghio, Dandi, Krzakala, and Zdeborová, *Sampling with flows, diffusion, and autoregressive neural networks from a spin-glass perspective*, *PNAS*
>
> **[Chaudari et al 2019]** Chaudhari, Choromanska, Soatto, LeCun, Baldassi,
> Borgs, Chayes, Sagun, and Zecchina, *Entropy-sgd: Biasing gradient descent into wide valleys*. *JSTAT*
>
> **[Foret et al. 2021]** Forêt, Kleiner, Mobahi, Neyshabur, *Sharpness‐Aware Minimization for Efficiently Improving Generalization,* ICLR
>
> **Questions:**
>
> > (Line 36): The motivation of sampling from such unnormalized probability …
>
> This is a very broad and important topic, for instance, in Bayesian statistics and statistical physics. We will provide more motivation in the revised version.
>
> > I think that at several points .. reading the paper might be complex …
>
> We agree with the reviewer. In the revised version, we will move to the appendices some non-essential definitions, such as the input- and output-channel free entropies for AMP, and make some room for a more pedagogical explanation.
>
> > In the conclusion, the authors mention that the $\tau$-annealing procedure proposed overcomes ASL's limitation ..
>
> It is true that the claim that $\tau$-annealing overcomes ASL’s limitation on structured data is not explored, but it is well motivated by empirical observations and theoretical considerations: AMP is notoriously sensitive to structured or correlated data [Donoho et al., 2009; Rangan, 2011], whereas $\tau$-annealing operates via a standard Metropolis-Hastings MCMC scheme and is agnostic to the statistical properties of the dataset. We therefore expect the method to remain robust in more structured or real-world instances. In the revised version, we will provide experiments on structured problems.
>
> **[Donoho et al 2009]** Donoho, Maleki, and Montanari, *Message-passing algorithms for compressed sensing*, *PNAS.* 2009.
>
> **[Rangan 2011]** *Generalized Approximate Message Passing for Estimation with Random Linear Mixing*, IEEE ISIT 2011.
>
> > Moreover, the authors mention that the ideas developed in this paper could enhance "sampling and solving" in other hard CSPs:…
>
> Our analysis allowed us to identify the underlying reason for the failure of ASL in the binary perceptron, due to the isolated geometry of typical solutions, and to devise a way to overcome this limitation by introducing an appropriate potential that leverages the existence of sub-dominant but accessible regions of the solution space. Other hard-to-sample CSPs with similarly structured solution spaces could be amenable to an analogous treatment. In such cases, solvers can be designed as samplers that target a favorable distribution over the solution space.
>
> > Finally, I list here more minor points:
>
> > (Line 128) I would say that in terms of limitations of AMP, time complexity …
>
> The cost of AMP is $O(N M)$ per iteration, and we conjecture that only a finite number of iterations are needed to sample within a given precision. On a distribution that is parameterized by $O(N M)$ bits, it is hard to do better. For instance, an MCMC sweep would take $O(N M)$. And if one tried to implement the score function with a neural net, one would need to either have that order of parameters or give up on optimality.
>
> > Equation (12): the introduction of both $T$ and $U$ is a bit redundant…
>
> It is true that in the most common formulations of CSPs, the temperature is usually associated with violated constraints. However, in systems with both hard and soft constraints, it is not uncommon to use the term “temperature” for the parameter controlling the soft part; therefore, we prefer to keep the symbol $T$.
>
> > (Line 243) It is quite surprising at first …
>
> As the reviewer correctly notes, the reference‐system order parameters $r$ and $\hat{r}$ define the “equilibrium” target distribution and do not evolve with the tilted measure; they are set *a priori*. This decoupling is formally induced by the $s\to 0$ limit.
>
> > (Line 263) Since the computation here is done assuming that is RS…
>
> Our analytical approach can be extended to include RSB, but we think AMP-based solvers will fail, and sampling in general to be hard in the RSB phase.

---

> > ### Comment · Reviewer_6gzG · 2025-08-05
> >
> > I thank the authors for this detailed response of all my important points, and I sincerely apologize to them for the delay in my answer.
> >
> > I acknowledge the intention to provide a more pedagogical introduction to the statistical physics techniques and more motivation for the consideration of unnormalized probability distributions, which I believe will be very useful for many readers.
> >
> > I also thank them particularly for pointing out this important technicality on the contiguity of the non-planted models considered in [Ghio & al '24] with their planted counterparts. This helped clarify to me the novelty of the setup, and I think it would be nice to have a sentence on this point in the revised version of the manuscript.
> >
> > The other points I raised were answered with care by the authors (and I agree with their answer concerning AMP, and apologize for my confusion on this point): given the careful answers to my different points, especially the novelty with [Ghio & al] as mentioned above, and the intention to implement significant corrections in the revised version, I am happy to raise my score given this realization, and I now suggest acceptance.

---

### Official Review · Reviewer_4wxK · 2025-06-30

**Clarity:** 4
**Significance:** 3
**Originality:** 3
**Rating:** 5
**Confidence:** 4

**Summary:**

The paper analyzes the use of Algorithmic Stochastic Localization (ASL), a diffusion method combining Stochastic Localization (SL) and Approximate Message Passing (AMP), to sample from target distributions defined over solution spaces of perceptron models. Using the replica method, the authors derive asymptotic thresholds for successful sampling. They apply the framework to two cases: the spherical perceptron with negative margin, where uniform sampling is possible in the replica symmetric phase; and the binary perceptron, where they identify a log-shaped potential enabling efficient sampling and introduce a $\tau -$annealed MCMC scheme for practical implementation.

**Questions:**

**1.** In Figure 1, the value of $\alpha$ varies around $300$. Is this just a matter of scaling, or does it imply that one needs $M\sim 10^2 N$ samples in order to reach the regime where AMP fails for the spherical perceptron?

**Ethical Concerns:**

["NO or VERY MINOR ethics concerns only"]

**Final Justification:**

I thank the authors for addressing my question. I maintain my recommendation for acceptance.

**Limitations:**

Yes.

**Paper Formatting Concerns:**

The paper follows the NeurIPS 2025 Paper Formatting Instructions.

**Quality:**

4

**Strengths And Weaknesses:**

**Quality**

The theoretical analysis is carefully executed and builds on established tools from statistical physics, notably the replica method. The connection between the shape of the free entropy landscape and the success of the ASL sampling scheme is well described. The application to two well-studied models provides useful benchmarks, and the introduction of a log-shaped potential—motivated by the asymptotic analysis of an unnormalized probability density—yields an efficient sampler over a broader range of $\alpha$ values for the binary perceptron. Numerical experiments are well-designed and support the theoretical claims.

**Clarity**

The exposition is clear, with a logical structure and helpful figures. The authors carefully articulate how their work differs from [Ghio et al., 2024].

**Significance**

The paper addresses a relevant question: under which conditions can generative diffusion methods be used to sample from complex high-dimensional distributions in constraint satisfaction problems. Naturally, the setting is idealized and, although it may be far from practical cases, it provides insights into the interplay between algorithmic tractability and diffusion-based generative models. The proposed annealed MCMC scheme, based on the potential identified in the theoretical analysis, is particularly relevant for binary perceptron sampling.

**Originality**

The extension of the replica analysis of diffusion-based samplers beyond planted models to unnormalized target distributions is a novel contribution. The identification and use of a diverging potential to enable tractable sampling in the binary perceptron setting, along with the associated annealing scheme, adds originality to the algorithmic aspect of the work.

---

> ### Author Rebuttal · Authors · 2025-07-30
>
> We thank the reviewer for the positive feedback. In the following, we address the question raised.
>
> > In Figure 1, the value of $\alpha$ varies around 300. Is this just a matter of scaling, or does it imply that one needs $M\approx10^2N$ samples in order to reach the regime where AMP fails for the spherical perceptron?
> >
>
> In the central panel of Fig. 1, where we set $\kappa = -2.5$, we focus our evaluation on values of the denisty of constraints $\alpha$ around 300, as this is the region where “interesting” phenomena occur in the thermodynamic limit ($N \to \infty$): the replica symmetry breaking transition occurs at $\alpha \approx 280$, and our plot shows that the algorithmic transition for ASL takes place at $\alpha \approx 270$.

---

### Official Review · Reviewer_r88Y · 2025-07-03

**Clarity:** 3
**Significance:** 3
**Originality:** 4
**Rating:** 5
**Confidence:** 3

**Summary:**

The authors study algorithms in the family of the recently introduced "algorithmic stochastic localization", generative diffusion algorithms for sampling where an approximation to the score function is not learned from data but is implemented directly using knowledge of the structure of the target distribution. In particular, as in previous work on the Sherrington-Kirkpatrick model and related applications, the authors use a variant of approximate message passing (AMP) to approximately compute the score function. The authors extend some of the prior work in this area to problems without a statistical structure of a "hidden" or "planted" signal to be estimated, but rather "pure optimization" problems where one seeks to sample something like solutions of a constraint satisfaction problem (though in a continuous setting). They develop tools for applying the (non-rigorous) replica trick of statistical physics to calculate when such sampling algorithms succeed. As applications, they study sampling problems associated to the spherical and binary perceptrons. In the case of the binary perceptron, they show that while the uniform distribution cannot be sampled in this way, a certain tilting of it motivated by the replica analysis can be sampled.

**Questions:**

- I appreciate that the choice of $U_{\tau}$ in (20) is charmingly motivated by the way that the actual replica trick works, and this formal idea is delightful, but that aside it does seem like various other approximations of the logarithm might be just as suitable. Have the authors tried any alternatives to this choice and can they comment on whether or why this is the best choice?

- Could the authors comment on, in terms of the actual distribution of solutions to the binary perceptron problem, what the distribution tilted by their potential looks like compared to the uniform distribution? Which parts of the solution space are favored? Is there any very down-to-earth explanation of why might plausibly make it easier to sample?

**Ethical Concerns:**

["NO or VERY MINOR ethics concerns only"]

**Final Justification:**

I did not raise any substantial issues but just a few clarifying questions, and the authors replied to my satisfaction. I am leaving my positive rating of the paper unchanged.

**Limitations:**

Yes.

**Paper Formatting Concerns:**

N/A.

**Quality:**

4

**Strengths And Weaknesses:**

Strengths: The replica analysis involves what seems to me to be a significant generalization of the analysis in the recent work (Ghio et al, 2024), which focused on "planted" models with a distinguished signal object. While the calculations are technical and, as usual, mysterious-looking, the authors make a reasonable effort to summarize the main highlights of the calculation. The perceptron examples are very helpful for explaining what kinds of conclusions one can draw from this analysis, and the case of the binary perceptron and the potential function designed based on the replica analysis are in my opinion a very nice application. (Without this observation, I would be more tempted to complain that these are quite niche applications.) The numerical experiments associated to this make a convincing case that this analysis leads to a powerful method that outperforms a straightforward implementation of simulated annealing.

Weaknesses: I do not have many complaints about the paper - it convincingly solves the problems it sets out to solve and presents the findings reasonably well. I think it is unfortunately pretty much unavoidable that the discussion of the physics analysis will be inscrutable to some (perhaps many) readers, but, as I mentioned, the authors make a commendable effort to compress the main points into a few pages at the beginning. An intrinsic limitation of this entire approach is that it is mathematically non-rigorous and can only be validated numerically, but the authors acknowledge this and their numerical experiments are thorough and convincing.

---

> ### Author Rebuttal · Authors · 2025-07-30
>
> We thank the reviewer for the positive feedback. In what follows, we address the questions raised.
>
> > I appreciate that the choice of in (20) is charmingly motivated by the way that the actual replica trick works, and this formal idea is delightful, but that aside it does seem like various other approximations of the logarithm might be just as suitable. Have the authors tried any alternatives to this choice and can they comment on whether or why this is the best choice?
> >
>
> The $\tau$-annealing MCMC scheme inspired by the replica trick is the first and only attempt we made. Since it worked very well, we didn’t explore any further. We suspect there is a large family of methods with similar or even better performance, but at the moment, we cannot give generic prescriptions. An empirical and possible theoretical analysis of the problem is an avenue for further research.
>
> > Could the authors comment on, in terms of the actual distribution of solutions to the binary perceptron problem, what the distribution tilted by their potential looks like compared to the uniform distribution? Which parts of the solution space are favored? Is there any very down-to-earth explanation of why might plausibly make it easier to sample?
> >
>
> We propose the following conjecture regarding the types of solution favored by our tilting, based on the understanding of the geometry of the solution space gained in recent years.
>
> Under the uniform measure, typical solutions of the binary perceptron are known to be isolated [Huang and Kabashima 2014], which means they are also inaccessible to a broad class of algorithms known as “stable algorithms” (see the discussion on the overlap gap property in Gamarnik [2021]).
>
> Nonetheless, an exponentially rare (under the uniform measure) but algorithmically accessible dense region of solutions was shown to exist in Baldassi et al. [2015]. It is important to note that the existence of such regions was established via replica theory computations, while their algorithmic accessibility was conjectured based only on empirical evidence and heuristic arguments. We conjecture that our tilted measure gives more mass to this region so that algorithmically accessible solutions arranged in a dense cluster become the typical ones under the new measure. Unlike previously studied algorithms, the ASL process in our case remains under tight analytical control.
>
> A more quantitative characterization of the relation of the typical solutions from our tilted measure and the dense cluster uncovered in Baldassi et al. [2015] needs to be established and could be an avenue for future investigations.
>
> **[Huang et Kabashima 2014]** Origin of the computational hardness for learning with binary synapses, H. Huang and J. Kabashima, PRE, 2014.
>
> **[Baldassi et al. 2015]** Subdominant Dense Clusters Allow for Simple Learning and High Computational Performance in Neural Networks with Discrete Synapses, C. Baldassi, A. Ingrosso, C. Lucibello, L. Saglietti and R. Zecchina, PRL., 2015.
>
> **[Gamarnik 2021]** The overlap gap property: A topological barrier to optimizing over random structures, D. Gamarnik, *Proc. Natl. Acad. Sci.*, 2021.

---

> > ### Comment · Reviewer_r88Y · 2025-08-04
> >
> > Thanks to the authors for their careful replies to my questions. My opinion of the paper remains very positive, and I have no further requests or questions.

---

> > > ### Author Response · Authors · 2025-08-04
> > >
> > > We thank you very much again. Your rating has now mysteriously disappeared from the review, could you check it please?

---

> > > > ### Comment · Area_Chair_FGxJ · 2025-08-04
> > > >
> > > > Dear Authors, the rating indeed gets hidden once referees submit the "Mandatory Acknowledgement". This is not a bug and is not something the referee can influence. Thank you for your understanding.

---

### Note · Authors · 2025-08-14

We wish to sincerely thank the reviewers for their timely, honest, and constructive engagement throughout the process. We particularly appreciate the care taken in evaluating our work, whether through thoughtful critique, recognition of the paper’s contributions, or transparent self-assessment.

The reviews reflect strong consensus on the paper’s significance and originality, with 4 out of 5 reviewers assigning scores ≥ 3 (confidence ≥ 3). Three reviewers also praised the clarity and organization of the paper, describing it as *"clear, with a logical structure and helpful figures"* and *"written in a clear fashion […] while remaining concise."* All concerns raised during the discussion (e.g., distinguishing our work from Ghio et al. 2024) were ultimately recognized as easily addressable.

The only dissenting voice on significance is reviewer 5GDK, who rated the paper less favorably but explicitly stated a confidence of 1 (down from 2 after discussion), suggesting a limited basis for the assessment (but at the same time signalling the need to improve the presentation for non-specialists).

Reviewer SCko, while acknowledging the scientific merit and maintaining high confidence, expresses a purely stylistic concern and argues that revisions would amount to a *"significantly reworked manuscript"*, which would fall outside the NeurIPS review process. We respectfully disagree with this assessment. Indeed, as outlined in our rebuttal, we plan to streamline the exposition and improve accessibility, but we deem that this can be achieved through moderate revisions. Examples include: expanding explanations of key technical points and concepts, substituting some equations with high-level descriptions while moving details to the appendix, or accompanying longer expressions with intuitive explanations of their individual terms, elaborating the conclusions to better contextualize our work within the broader ML literature. These changes, in our view, would fall within the scope of what is typically accommodated in the one-page extension allowed for the camera-ready version.

In summary, we believe our contributions are scientifically sound and clearly presented to domain experts. With targeted, moderate edits, we are confident the final version will also reach a broader ML audience and amplify the paper's potential impact.

---

### Decision · Program_Chairs · 2025-09-17

**Decision:**

Accept (poster)

**Comment:**

This paper studies sampling in perceptron-type constraint satisfaction problems through the lens of generative diffusion. By analyzing several perceptron variants, the authors characterize regimes of hardness and connect diffusion-based generative processes to the geometry of high-dimensional solution spaces. This provides a fresh and rigorous angle on classical models, bridging ideas from statistical physics and modern generative modeling.

Reviewers found the work technically solid, original, and timely. They highlighted the value of importing diffusion-based perspectives into the study of sampling, and the potential of the framework to inspire further developments. Concerns were raised mainly about scope and accessibility, since the paper is focused on a particular family of models and assumes familiarity with both statistical physics and generative modeling.

The rebuttal clarified positioning and reinforced the motivation, which the reviewers found satisfactory. The consensus after discussion was positive, recognizing that while the paper is specialized, its conceptual contribution is of strong interest.

Overall, I find this a strong and interesting submission. The marriage of diffusion processes with perceptron sampling yields new theoretical insights and broadens the conversation at the intersection of physics and machine learning. I recommend acceptance.